# Synergizing Deconfounding and Temporal Generalization For Time-series Counterfactual Outcome Estimation

**Yiling Liu**                                                                           *yiling.liu@duke.edu*
*Program in Computational Biology and Bioinformatics, Duke University*

**Juncheng Dong**                                                                   *juncheng.dong@duke.edu*
*Department of Electrical and Computer Engineering, Duke University*

**Chen Fu**                                                                                   *chenfu@meta.com*
*AI at Meta, Meta Platforms, Inc.*

**Wei Shi**                                                                               *weishi0079@meta.com*
*AI at Meta, Meta Platforms, Inc.*

**Ziyang Jiang**                                                                         *jzy95310@meta.com*
*AI at Meta, Meta Platforms, Inc.*

**Qi Xu**                                                                                 *xuqi0511@gmail.com*
*AI at Meta, Meta Platforms, Inc.*

**Zhigang Hua**                                                                             *zhua@meta.com*
*AI at Meta, Meta Platforms, Inc.*

**David Carlson**[†]                                                             *david.carlson@duke.edu*
*Department of Civil and Environmental Engineering*
*Department of Biostatistics and Bioinformatics*
*Department of Computer Science*
*Duke University*

**Reviewed on OpenReview:** *https://openreview.net/forum?id=xuJH3BJiNu*

## Abstract

Estimating *counterfactual* outcomes from time-series observations is crucial for effective decision-making, e.g. when to administer a life-saving treatment, yet remains significantly challenging because (*i*) the counterfactual trajectory is never observed and (*ii*) confounders evolve with time and distort estimation at every step. To address these challenges, we propose a novel framework that ***synergistically integrates*** two complementary approaches: ***Sub-treatment Group Alignment*** (SGA) and ***Random Temporal Masking*** (RTM). Instead of the coarse practice of aligning marginal distributions of the treatments in latent space, SGA uses iterative *treatment-agnostic* clustering to identify fine-grained *sub-treatment groups*. Aligning these fine-grained groups achieves improved distributional matching, thus leading to more effective deconfounding. We theoretically demonstrate that SGA optimizes a tighter upper bound on counterfactual risk up to an additive constant term, and empirically verify its deconfounding efficacy. RTM promotes temporal generalization by randomly replaces input covariates with Gaussian noises during training. This encourages the model to rely less on potentially noisy or spuriously correlated covariates at the current step and more on stable historical patterns, thereby improving its ability to generalize across time and better preserve underlying *causal relationships*. Our experiments demonstrate that while applying SGA and RTM individually improves counterfactual outcome estimation, their

---

[†]Corresponding author.

synergistic combination consistently achieves state-of-the-art performance. This success comes from their distinct yet complementary roles: RTM enhances temporal generalization and robustness across time steps, while SGA improves deconfounding at each specific time point.

## 1 Introduction

Estimating causal effects from time series data is a pivotal task in many fields such as healthcare, politics, and economics (Bisgaard & Kulahci, 2011; Freeman, 1983; Morid et al., 2023). This fundamentally requires the ability to estimate counterfactual outcomes: what would have happened under different interventions over time. For example, in the treatment of *Ductal Carcinoma In Situ*, accurately estimating counterfactual outcomes is crucial for determining the timing of surgical intervention: if surgery is too late, the cancer may progress to an invasive stage; if conducted too early, the procedure may be unnecessarily invasive (Grimm et al., 2022).

Motivated by this, we explore counterfactual outcome estimation in time series from observational data. The success of causal inference in time series relies on **effective reduction of time-dependent confounding**. However, this task is challenging, primarily because of the unobservability of counterfactual outcomes and the time-varying confounding in time series. A well-established approach for reducing confounding in *static* causal inference is to minimize an upper bound on the counterfactual estimation error (Johansson et al., 2016; 2022; Li & Fu, 2017; Yao et al., 2018), which can be decomposed into two key components: (*i*) the *factual loss* and (*ii*) the *statistical discrepancy between treatment and control groups* in the learned representation space. Algorithmically, these methods simultaneously (*i*) minimize the prediction error of the factual outcomes and (*ii*) align the treatment groups in the latent space. By ensuring that the representations of multiple treatment groups are brought closer together, they provably reduce the bias introduced by confounders (Johansson et al., 2022). Building on this idea to reduce confounding for time series, existing approaches aim to learn representations that remain invariant to the treatment assignment **at each time step** (Bica et al., 2020; Melnychuk et al., 2022). However, in practice, they typically result in optimizing relatively **loose** upper bounds on the counterfactual error at individual time steps (Arjovsky & Bottou, 2017) with adversarial training. Moreover, models may **over-rely on contemporaneous information** at each time step, potentially hindering their ability to learn stable long-term patterns across time points (Ghouse et al., 2024). This can lead to **compromised generalization** for estimating long-term effects, as errors can propagate and compound over time.

To address these challenges, we introduce our framework with two novel approaches:

- ❖ *Sub-treatment Group Alignment (SGA)*, which improves deconfounding at each individual time point by identifying and subsequently aligning *sub-treatment groups*.

- ✯ *Random Temporal Masking (RTM)* promotes temporal generalization and robust learning from historical patterns by randomly masking covariates at selected time points with Gaussian noise.

**Sub-treatment Group Alignment (SGA).** SGA first identifies **sub-treatment groups in the representation space** through *treatment-agnostic* clustering at each timestep, and then subsequently aligning the corresponding sub-groups across different treatment groups (see Figure 1). In Section 4, we establish that such sub-group alignment indeed leads to a tighter bound on the counterfactual estimation error up to an additive constant term. This more fine-grained alignment enables **improved matching** of treatment groups, thus allowing us to **reduce the estimation error** more effectively than existing methods.

**Random Temporal Masking (RTM).** While SGA addresses confounding at **individual** time points, RTM enhances the model's ability to generalize **across** time series. Inspired by masked language modeling, RTM uses random covariate masking, where input covariates at randomly selected time points are replaced with Gaussian noise during training. There are multiple perspectives to understand the benefits of RTM: (*i*) When input covariates at certain time points are masked, the models must extract useful information from past observations to predict future factual outcomes. In other words, we encourage the model to **focus on the causal relationships that span across time**, leading to better counterfactual predictions. (*ii*) RTM

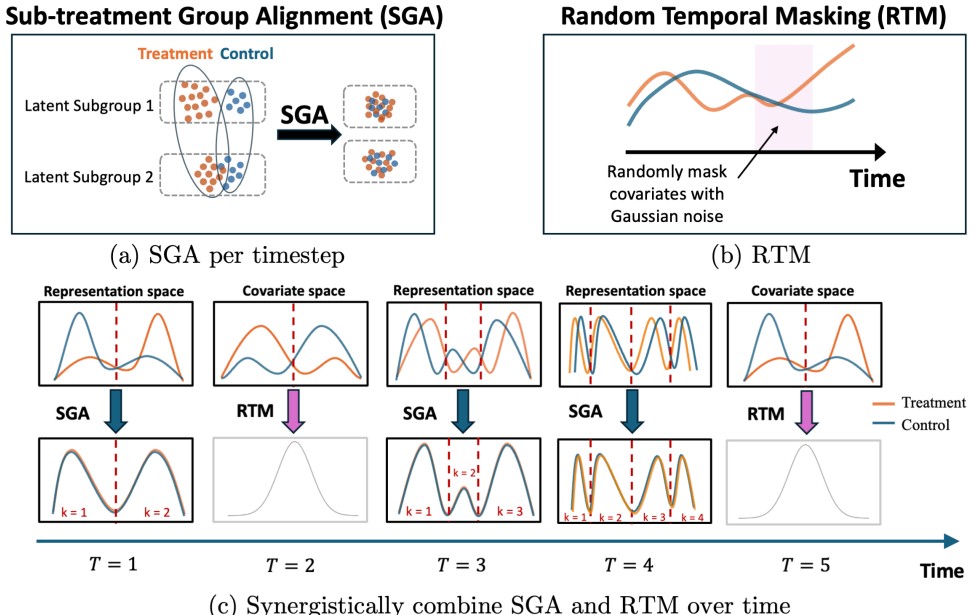

(a) SGA per timestep  (b) RTM

(c) Synergistically combine SGA and RTM over time

Figure 1: **Conceptual overview of SGA and RTM.** **(a)** SGA identifies and aligns fine-grained sub-treatment groups at each timestep to improve deconfounding. **(b)** RTM forces the model to leverage historical patterns and enhancing temporal generalization. **(c)** SGA and RTM are synergistically combined to improve counterfactual outcome estimation. Here, $k$ denotes sub-treatment group index.

can prevent model from becoming overly reliant on information from current time points, thus **reducing overfitting to the factual distribution**.

**Empirical Validation.** Our framework is **broadly applicable** and designed for **straightforward integration** into existing representation learning-based frameworks for time series counterfactual estimation. We empirically validate our framework through comprehensive experiments on synthetic and semi-synthetic datasets, and it consistently demonstrates state-of-the-art (SOTA) performance.

**Organization.** We first formally define the problem in Section 2 and review related works in Section 3. Then in Section 4, we theoretically establish how sub-treatment group alignment achieves improved deconfounding, thus motivating our SGA approach. In Section 5, we present our framework with SGA and RTM. Finally, experimental results in Section 6 show that applying SGA and RTM individually enhances performance, and that their *synergistic combination* achieves SOTA results.

## 2 Problem Setup

**Notations.** We use upper-case letters (e.g., $A, Y$) for scalar random variables and lower-case letters (e.g., $a, y$) for their corresponding realizations. Multi-dimensional random variables and realizations are denoted using bold fonts (e.g., $\boldsymbol{X}$ and $\boldsymbol{x}$).

**Observational Data.** We consider a dataset containing $N$ samples following conventional setups (Bica et al., 2020; Li et al., 2020; Melnychuk et al., 2022). Observations are recorded over $T$ time steps, i.e., $t = 1, ..., T$. At each time $t$, a discrete treatment $A_t \in \mathcal{A} = \{a_0, a_1, ..., a_{|\mathcal{A}|-1}\}$ is assigned. Thus, for each sample $i \in \{1, 2, ..., N\}$, we observe time-varying covariates $\mathbf{X}_t^{(i)} \in \mathbb{R}^d$, factual treatment $A_t^{(i)}$, and outcome $Y_t^{(i)}$ of the factual treatment.

We use the following notation to represent the history up to time step $t$ for each unit $i$:

$$\bar{\mathbf{H}}_t^{(i)} = \{\bar{\mathbf{X}}_t^{(i)}, \bar{\mathbf{Y}}_t^{(i)}, \bar{\mathbf{A}}_{t-1}^{(i)}, \mathbf{V}^{(i)}\}$$

where $\bar{\mathbf{X}}_t^{(i)} = \{\mathbf{X}_s^{(i)} : s \leq t\}$ denotes the sequence of time-varying covariates up to time $t$, $\bar{\mathbf{Y}}_t^{(i)} = \{Y_s^{(i)} : s \leq t\}$ represents the sequence of observed outcomes up to time $t$, $\bar{\mathbf{A}}_{t-1}^{(i)} = \{A_s^{(i)} : s \leq t-1\}$ is the sequence of treatments up to time $t-1$, and $\mathbf{V}^{(i)} \in \mathbb{R}^p$ denotes the static covariates.

**Potential Outcomes Framework with Time-Varying Treatments and Outcomes.** Building on the potential outcomes framework (Rosenbaum & Rubin, 1983; Rubin, 2005), we extend these assumptions to accommodate time-varying treatments and outcomes, following Robins & Hernan (2008).

**Assumption 2.1** (Consistency). If $\bar{\mathbf{A}}_t = \bar{\mathbf{a}}_t$ is a given sequence of treatments for some patient, then $\mathbf{Y}_{t+1}[\bar{\mathbf{a}}_t] = \mathbf{Y}_{t+1}$. This means an individual's potential outcome under the observed treatment history is the observed outcome.

**Assumption 2.2** (Sequential Positivity). Positivity states that there is a non-zero probability of receiving any of the counterfactual treatments. It can be expressed as $0 < P(\mathbf{A}_t = \mathbf{a}_t | \bar{\mathbf{H}}_t = \bar{\mathbf{h}}_t) < 1$, if $P(\bar{\mathbf{H}}_t = \bar{\mathbf{h}}_t) > 0$.

**Assumption 2.3** (Sequential Ignorability). There is no unobserved confounding of treatment at any time and any future outcome. This can be expressed as $\mathbf{A}_t \perp\!\!\!\perp \mathbf{Y}_{t+1}[\mathbf{a}_t] | \bar{\mathbf{H}}_t, \forall\, \mathbf{a}_t \in \mathcal{A}$.

Using assumptions 2.1–2.3, Robins (1986) establishes the sufficient conditions for identifiability through G-computation, ensuring that causal effects can be appropriately identified.

**Causal Directed Acyclic Graphs (DAGs).** Figure 2 provides a visual summary of the assumed data-generating mechanisms in both static and time-series settings. These diagrams illustrate the causal relationships among covariates, treatments, and outcomes, and highlight the challenges introduced by time-varying confounding.

**Objective.** Given the history up to current time $t$ and assuming a specific treatment sequence $\mathbf{a}_{t:t+\tau-1}^{(i)}$ from time $t$ to $t + \tau - 1$ applied to sample $i$, **our goal is to estimate**, for each unit $i$, **the future outcome at time step** $t + \tau$. To ensure that these counterfactual outcomes are identifiable, we rely on the potential outcomes framework outlined above. Specifically, we estimate:

$$\mathbb{E}[Y_{t+\tau}^{(i)}(\mathbf{a}_{t:t+\tau-1}^{(i)}) \,|\, \bar{\mathbf{H}}_t^{(i)}], \tag{1}$$

where $Y_{t+\tau}^{(i)}(\mathbf{a}_{t:t+\tau-1}^{(i)})$ is the potential outcome at $t + \tau$ for unit $i$ with treatment sequence $\mathbf{a}_{t:t+\tau-1}^{(i)}$.

## 3 Related Work

We review the **most** relevant work below and provide a **comprehensive** discussion in Appendix B.

**Estimating Counterfactual Outcomes Over Time.** Estimating counterfactual outcomes in time-series is challenging due to time-varying confounders. Traditional methods such as G-computation and marginal structural models (Hernán et al., 2001; Robins, 1986; Robins & Hernan, 2008; Robins et al., 2000; Xu et al., 2016) often lack flexibility for complex datasets and rely on strong assumptions. To address these limitations, researchers have developed models that build on the potential outcomes framework initially proposed by Rubin (1978) and extended to time series by Robins & Hernan (2008). Notable among recent methods are Recurrent Marginal Structural Networks (RMSNs) (Lim, 2018), G-Net (Li et al., 2020), Counterfactual Recurrent Networks (CRN) (Bica et al., 2020), and the Causal Transformer (CT) (Melnychuk et al., 2022), which use approaches such as propensity networks and adversarial learning to mitigate the effects of time-varying confounding. However, practical challenges with adversarial training can affect the stability of causal effect estimations. Specifically, training adversarial networks can be challenging due to issues such as mode collapse and oscillations (Liang et al., 2018). Additionally, adversarial training minimizes the Jensen-Shannon divergence (JSD) only when the discriminator is optimal (Arjovsky & Bottou, 2017), which may not always be achievable in practice; even when the discrminator is optimal, using JSD optimizing relatively loose upper bounds on the counterfactual error. To address these challenges, we propose using the Wasserstein-1 distance and provides stronger theoretical guarantees (Mansour et al., 2012; Redko et al., 2017).

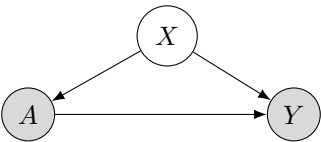

(a) Static (Non-Time-Series) Scenario

*In the static setting, $A$ denotes treatment assignment, $X$ the observed covariates, and $Y$ the outcome. The covariates act as confounders affecting both treatment and outcome.*

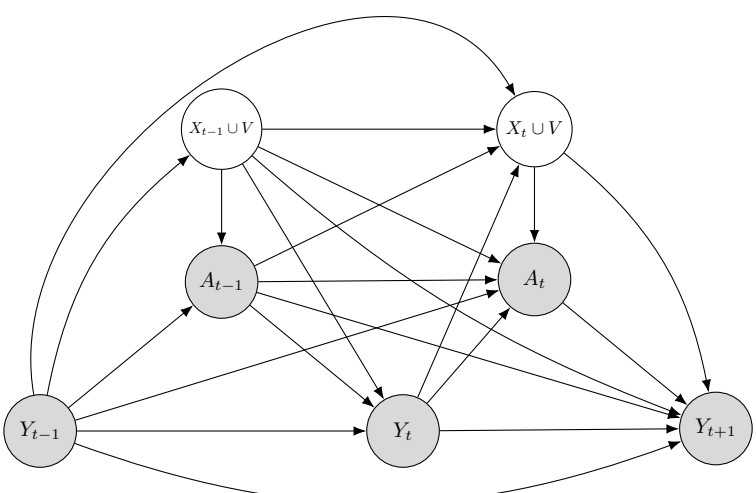

(b) Time-Series Scenario

*In the time-series setting, $A_t$ denotes treatment at time $t$, $X_t \cup V$ represents observed time-varying and static covariates, and $Y_t$ denotes the outcome at time $t$. Past treatments and outcomes influence future covariates, treatments, and outcomes, inducing time-varying confounding.*

Figure 2: **Causal Directed Acyclic Graphs (DAGs) Illustrating Causal Relationships.** (a) Static setting. (b) Time-series setting with dynamic treatment effects and time-varying confounding.

**Masked Language Modeling.** Masked language modeling (MLM) is a common self-supervised pre-training technique for large language models. It operates by randomly masking certain words or tokens in the input, with the model trained to predict the masked tokens. BERT (Devlin, 2018) is the most well-known model that uses this technique. Recent studies have also demonstrated the effectiveness of MLM in enhancing generalization across sequence-based tasks. For example, Chaudhary et al. (2020) shows that when combined with cross-lingual dictionaries, MLM improves predictions for the original masked word and also generalizes to its cross-lingual synonyms. Inspired by the success of masking strategies in language models, we introduce Random Temporal Masking (RTM) for time-series data. Unlike MLM, which focuses on predicting the masked inputs, RTM encourages the model to focus on information that is crucial for both the current time point and future time points, preserve causal information, and reduce the risk of overfitting to factual outcomes.

## 4 Theoretical Motivation for Sub-treatment Group Alignment

In this section, we provide a theoretical motivation for our proposed Sub-treatment Group Alignment (SGA) method, rigorously illustrating that **aligning sub-treatment groups in the latent space leads to more effective deconfounding in estimating counterfactual outcomes** over time series.

**From Static to Time Series.** By aligning the corresponding sub-treatment groups, **SGA achieves improved alignment** and thus more effective deconfounding. Since existing time series methods often apply alignment independently at each time step (Bica et al., 2020; Melnychuk et al., 2022), demonstrating SGA's superiority in a static context provides a strong foundation for its benefits in time-series. In other words, achieving better deconfounding at *individual time steps* will consequently lead to more effective deconfounding *over the entire time series.*

**Section Organization.** Given the rationale that time-series alignment can be decomposed into a collection of static sub-problems, **it is sufficient to consider static settings**. Thus, in Section 4.1 we briefly review (*i*) representation learning-based models that use alignment (*ii*) why alignment mitigates confounder bias in the static setting. Subsequently, in Section 4.2, we theoretically establish that SGA indeed improves alignment in static settings. This implies that **integrating SGA into existing time-series frameworks can improve deconfounding at each step**, leading to overall improvements.

## 4.1 Alignment for Static Setting

Since there is only one time step $t = 1$ in the static setting, we will omit all notations about the time step for clarity. To establish the theoretical bounds, we first define the necessary mathematical components, including the distance metric and the objective functions. Due to space constraints, we defer the discussion of SGA's computational cost to Appendix F.

**Wasserstein Distance.** The Kantorovich-Rubenstein dual representation of the Wasserstein-1 distance (Villani, 2009) between two distributions $p_\Phi^0$ and $p_\Phi^1$ is defined as:

$$W_1(p_\Phi^0, p_\Phi^1) = \sup_{\|f\|_L \leq 1} \mathbb{E}_{x \sim p_\Phi^0}[f(x)] - \mathbb{E}_{x \sim p_\Phi^1}[f(x)],$$

where the supremum is over the set of 1-Lipschitz functions (all Lipschitz functions $f$ with Lipschitz constant $L \leq 1$). For notational simplicity, we use $D(X_1, X_2)$ to denote a distance between the distributions of any pair of random variables $X_1$ and $X_2$.

**Lipschitz Functions.** For some $K \geq 0$, the set of $K$-Lipschitz functions consists of all functions $f$ satisfying:

$$\|f(x) - f(x')\| \leq K\|x - x'\|, \quad \forall x, x' \in \mathcal{X}.$$

We assume that the hypothesis class $\mathbb{H}$ is a subset of $\lambda_H$-Lipschitz functions for some constant $\lambda_H > 0$, and that the true labeling functions are $\lambda$-Lipschitz for some $\lambda > 0$.

**Factual and Counterfactual Losses.** Following Shalit et al. (2017), let $\Phi : \mathcal{X} \to \mathcal{R}$ be a representation function and $h : \mathcal{R} \times \{0, 1\} \to \mathcal{Y}$ be a hypothesis defined over the representation space. Let $Y_a \in \mathcal{Y}$ denote the potential outcome under treatment assignment $a \in \{0, 1\}$. We define the expected loss for unit-treatment pair $(x, a)$ as:

$$\ell_{h,\Phi}(x, a) = \int_{\mathcal{Y}} L\big(Y_a, h(\Phi(x), a)\big) \, p(Y_a \mid x) \, dY_a,$$

where $L : \mathcal{Y} \times \mathcal{Y} \to \mathbb{R}_+$ is a loss function. The expected factual and counterfactual losses are defined as:

$$\epsilon_F(h, \Phi) = \int_{\mathcal{X} \times \{0,1\}} \ell_{h,\Phi}(x, a) \, p(x, a) \, dx da,$$

$$\epsilon_{CF}(h, \Phi) = \int_{\mathcal{X} \times \{0,1\}} \ell_{h,\Phi}(x, a) \, p(x, 1 - a) \, dx da,$$

where $p(x, a)$ denotes the joint distribution over $\mathcal{X} \times \{0, 1\}$.

Let $u = p(a = 1)$ with $0 < u < 1$. We additionally define the reweighted factual loss:

$$\epsilon_F^\star(h, \Phi) = (1 - u) \, \epsilon_F^{a=1}(h, \Phi) + u \, \epsilon_F^{a=0}(h, \Phi),$$

which symmetrizes the factual error across treatment groups and will be used in the generalization bound.

**Representation Learning-based Models.** With these definitions established, we have $h(\Phi(x), a)$ as a predictor for an individual $x$'s potential outcome under treatment assignment $a$. The goal is to find a pair of $(h, \Phi)$ that optimizes both the *factual loss* $\epsilon_F^\star(h, \Phi)$ and *counterfactual loss* $\epsilon_{CF}(h, \Phi)$. Note that low factual and counterfactual losses are both necessary and sufficient conditions for accurate potential outcome prediction (Aloui et al., 2023).

**Counterfactual Error Estimation.** However, the **counterfactual loss $\epsilon_{CF}(h, \Phi)$ cannot be directly optimized** because the counterfactual outcomes are not observable in real-world scenarios. To this end, a group of well-established approaches minimize **upper bounds** of $\epsilon_{CF}(h, \Phi)$. These approaches are mainly based on the following result from Shalit et al. (2017), restated here.

**Theorem 4.1** (Simplified Lemma A8 from Shalit et al. (2017), complete version provided in Appendix C.12.). *Let $\Phi : \mathcal{X} \to \mathcal{R}$ be a one-to-one and Jacobian-normalized representation function. Let $h : R \times \{0, 1\} \to Y$ be a hypothesis with Lipschitz constant:*

$$\epsilon_{CF}(h, \Phi) \leq \epsilon_F^\star(h, \Phi) + 2 \cdot B_\Phi \cdot W_1(p_\Phi^0, p_\Phi^1), \tag{2}$$

*where $B_\Phi$ is a constant and $p_\Phi^a$ is the distribution of the random variable $\Phi(X)$ conditioned on $A = a$, that is, representations for individuals receiving treatment $a \in \{0, 1\}$.*

**Motivation for Alignment.** This theorem implies that a model $(\Phi, h)$ has low counterfactual error if *(a)* it has **low factual error** (which can be easily achieved by minimizing the prediction error on the observational data) and *(b)* the covariates of individuals from distinct treatment groups are **statistically similar to each other in the latent (representation) space**. Motivated by these, representation learning-based methods aim to align the treatment and control groups in the latent space while minimizing the factual error. In particular, successful alignment and low factual error guarantee a small value for the upper bound in Equation (2), implying the model has low counterfactual error. However, in practice, **the error bound may be loose**, leaving the model performance suboptimal (Arjovsky & Bottou, 2017).

## 4.2 Benefits of Sub-treatment Group Alignment

To this end, we propose to use the **sub-treatment group structures** to achieve **tighter** counterfactual error bound up to an additive constant term, thus supporting **more effective** alignment.

**Sub-treatment Groups.** We hypothesize that each treatment group is a mixture of $K$ sub-treatment groups in the latent space, and that **the sub-treatment groups across different treatment groups correspond to one another**. For example, in medical studies, patients may naturally form sub-groups **before** the beginning of experiments, based on latent variables such as demographic characteristics or genetic factors. Consider a scenario where patients are sub-grouped according to age (e.g., children, adults, seniors), gender, or genetic markers that influence their response to treatment. Even though these patients receive different treatments, the underlying characteristics defining the sub-groups are consistent across treatment groups. By aligning these corresponding sub-groups in the latent space, we can more effectively account for **hidden confounders** like genetic predispositions or socio-demographic factors, leading to more accurate estimation of treatment effects.

Recall $p_\Phi^a$ is the distribution of representations for individuals receiving treatment $a \in \{0, 1\}$, thus

$$p_\Phi^0 = \sum_{k=1}^K w_k^0 P_{\Phi,k}^0, \quad p_\Phi^1 = \sum_{k=1}^K w_k^1 P_{\Phi,k}^1,$$

where for $a \in \{0, 1\}$, $w_k^a$ represents the proportion of the $k$-th sub-group in treatment group $a$, and $P_{\Phi,k}^a$ is the distribution of representations of individuals in the $k$-th sub-group under treatment $a$.

**Intuition of Sub-treatment Group Alignment (SGA).** In many real-world settings, each treatment group is not homogeneous but instead forms a mixture of latent sub-populations. Marginal alignment minimizes the Wasserstein distance between the overall treatment distributions, but this coarse alignment can still permit substantial mismatch between corresponding latent sub-populations. In contrast, SGA decomposes each treatment distribution into sub-treatment components and aligns them pairwise. Intuitively,

instead of forcing two heterogeneous clouds of representations to overlap globally, SGA aligns them locally at the sub-population level, leading to more effective deconfounding.

**Sub-treatment Group Alignment (SGA).** SGA has the following alignment objective:

$$\sum_{k=1}^{K} w_k^1 W_1(P_{\Phi,k}^0, P_{\Phi,k}^1). \tag{3}$$

In particular, SGA minimizes the **weighted sum** of the Wasserstein distances between these **corresponding sub-treatment groups**. By aligning on a sub-treatment group level, SGA achieves more refined alignment. Indeed, motivated by the generalization bound in the field of *domain adaptation* (Liu et al., 2025), we next prove in Theorem 4.2 that SGA optimizes a finer-grained conditional objective that is theoretically tighter than marginal alignment (Theorem 4.1) up to an additive constant term, while empirically yielding a strictly tighter constraint on counterfactual risk in practice.

**Theorem 4.2** (SGA Improves Generalization Bounds). *Under the following assumptions:*

***A1.*** *For all $k$, the sub-distributions $P_{\Phi,k}^0$ and $P_{\Phi,k}^1$ are Gaussian distributions with means $m_k^0$ and $m_k^1$, and covariances $\Sigma_k^0$ and $\Sigma_k^1$, respectively. The distance between corresponding sub-distributions is less than or equal to the distance between non-corresponding sub-distributions, i.e., $W_1(P_{\Phi,k}^0, P_{\Phi,k}^1) \leq W_1(P_{\Phi,k}^0, P_{\Phi,k'}^1)$ for $k \neq k'$.*

***A2.*** *The sub-treatment group weights are identical across treatment groups, i.e., $w_k^1 = w_k^0$ for all $k \in [K]$. Moreover, there exists a small constant $\epsilon > 0$, such that $\max_{k \in [K]}(tr(\Sigma_k^0)) \leq \epsilon$ and $\max_{k \in [K]}(tr(\Sigma_k^1)) \leq \epsilon$.*

*The following inequalities hold:*

$$\epsilon_{CF}(h, \Phi) \leq \epsilon_F^\star(h, \Phi) + 2B_\Phi(\sum_{k=1}^{K} w_k^1 W_1(P_{\Phi,k}^0, P_{\Phi,k}^1))$$
$$\sum_{k=1}^{K} w_k^1 W_1(P_{\Phi,k}^0, P_{\Phi,k}^1) \leq W_1(p_\Phi^0, p_\Phi^1) + \delta_c,$$

*where $B_\Phi$ is the same constant in Theorem 4.1 and $\delta_c$ is $4\sqrt{\epsilon}$.*

See Appendix C for proof of Theorem 4.2 and detailed assumption statements and interpretations.

*Remark* 4.3. Because the identification of the sub-treatment group is *treatment-agnostic*, each unit's subgroup index is determined *solely* by the representation $\Phi(x)$, independent of the treatment label $A$. Consequently, whenever treatment assignment is randomized or conditionally ignorable given the representation ($A \perp k \mid \Phi(x)$), the probability of falling into cluster $k$ is identical across arms: $\Pr(k|A{=}0) = \Pr(k|A{=}1) = w_k$, thus the use of $w_k$, $w_k^0$ and $w_k^1$ is interchangeable.

*Remark* 4.4. Theorem 4.2 establishes that SGA optimizes an upper bound that is theoretically tighter than the original bound in Theorem 4.1 up to an additive constant term $\delta_c$. Equivalently, the SGA bound can be looser than the marginal one by at most $\delta_c$, so its tightness is controlled by this additive constant. Crucially, however, $\delta_c$ is small in practice: as shown in Appendix E.1.3, the SGA alignment loss consistently attains strictly lower discrepancy values than marginal alignment across varying confounding levels. This empirical reduction in the Wasserstein-based discrepancy term directly supports the claim that SGA achieves a tighter practical bound and leads to improved counterfactual prediction accuracy.

*Remark* 4.5. Theorem 4.2 uses the assumptions regarding the distributional properties of sub-treatment groups in the learned representation space $\mathcal{R}$. Specifically, Assumption **A1** posits that sub-distributions $P_{\Phi,k}^0$ and $P_{\Phi,k}^1$ (for treatment arms 0, 1) are Gaussian and that $W_1(P_{\Phi,k}^0, P_{\Phi,k}^1) \leq W_1(P_{\Phi,k}^0, P_{\Phi,k'}^1)$ for $k \neq k'$. Assumption **A2** states that the trace of their covariances is bounded by a small $\epsilon$. We justify the validity of these assumptions below in Appendix C.

# 5 Framework

We propose a framework that ***synergistically integrates*** our two novel approaches: Sub-treatment Group Alignment (SGA) and Random Temporal Masking (RTM). Figure 3 illustrates the general architecture of our framework.

**Model Architecture.** A key feature of our framework is its versatility; it is not restricted to a specific architecture and can be integrated with various representation-based methods for time series causal inference. Typically, such methods consist of a *time series encoder* $\phi_E$ (parameterized by $\theta_E$) that learns representations from input time series data, and an *outcome regressor* $f_Y$ (parameterized by $\theta_Y$) that predicts outcomes at the next time point. We note that the encoder $\phi_E$ can be instantiated with any sequence model architecture, such as RNNs, LSTMs (Hochreiter, 1997), or transformers (Vaswani, 2017). In Section 6, we demonstrate this flexibility by integrating our approaches with two well-established methods for time series causal inference: *Causal Transformer* (**CT**) (Melnychuk et al., 2022) and *Counterfactual Recurrent Networks* (**CRN**) (Bica et al., 2020).

**Random Temporal Masking (RTM)** *is applied to the observational data before the training of models.* To implement RTM, we mask covariates at a set of randomly selected time steps by replacing them with Gaussian noise. The model is subsequently trained to predict the outcomes despite these masked covariates, encouraging it to focus on causal information that is robust over time. RTM also reduces the risk of overfitting to factual outcomes at those selected time steps because, after masking, the covariates at current time is independent of the outcome. *This is particularly helpful when the potential outcomes at the current time steps are strongly correlated with current covariates* because under these scenarios the models are inclined to heavily rely on current covariates, thus overfitting to the factual distribution.

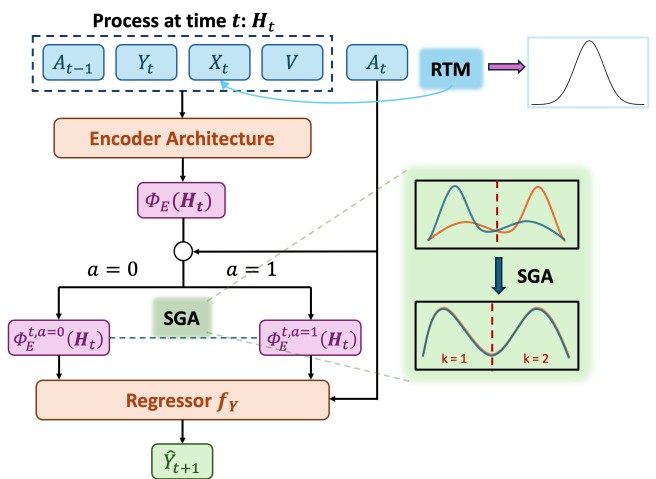

Figure 3: **Overview of SGA & RTM** at each timepoint. For simplicity, we show a binary treatment scenario.

Importantly, RTM does not explicitly model temporal dynamics; rather, it functions as a training-time regularization mechanism. By randomly masking covariates at selected time steps with Gaussian noise, RTM reduces the availability of contemporaneous information during training. Consequently, the model must rely more heavily on historical observations when predicting future outcomes. This encourages the learning of representations that capture dependencies stable across time rather than spurious correlations localized to a single time step.

In settings where current covariates are strongly correlated with factual outcomes, models without RTM may overfit to these contemporaneous signals, potentially impairing long-horizon counterfactual prediction. RTM mitigates this tendency by regularizing the hypothesis space and promoting temporal robustness. Empirically, this behavior is reflected in improved long-horizon performance and increased attention to past time steps as shown in Sec 6.5.

**Objective Function.** At each time step $t$, our framework optimizes the objective

$$\min_{\theta_Y, \theta_E} L_Y^t(\theta_Y, \theta_E) + \lambda L_D^t(\theta_E),$$

where $L_Y^t$ represents the **factual outcome loss** and $L_D^t$ denotes the **SGA loss** calculated with the Wasserstein-1 distance, balanced by $\lambda$. Details are provided below.

**Factual Outcome Loss.** At each time step $t$, the model learns to predict the observed outcomes, conditioned on $\mathbf{H}_t^{(i)}$ which contains the information from previous steps and the current covariates, by optimizing

$$L_Y^t(\theta_Y, \theta_E) = \frac{1}{N} \sum_{i=1}^N (\ell(y_i^{t+1}, \hat{y}_i^{t+1})),$$

where $\hat{y}_i^{t+1} = f_Y\left(\phi_E\left(\mathbf{H}_t^{(i)}, A_t^{(i)}\right)\right)$ and $\ell(\cdot, \cdot)$ denotes the loss function (e.g., mean squared error).

**SGA Loss.** Motivated by Section 4, our framework aligns the sub-treatment groups across distinct treatment groups. To this end, at each time step $t$ and for each treatment group $a$, we use Gaussian Mixture Models (GMMs) to cluster the individuals' features in the representation space into $K$ sub-treatment groups. Let the random variable $\phi_E^{t,a,k}(\mathbf{H}_t)$ denote the representations of samples in the $k$-th sub-group of treatment group $a$ at time step $t$.

To accommodate the applications with **multiple treatment groups** (more than two), for each time step $t$ and each corresponding sub-treatment group, we align the sub-treatment groups with **the uniform mixtures of them**. That is, for all the $k$-th sub-treatment groups in all $|\mathcal{A}|$ treatment groups where $|\mathcal{A}|$ is the total number of treatments, we first create a mixture of them with uniform weights and align all of them with the uniform mixture. Note that by triangle inequality this is a **sufficient condition** to align multiple groups well. Specifically, the SGA loss is defined as:

$$L_D^t(\theta_E) = \sum_{k=1}^K \sum_{a \in \mathcal{A}} w_k^{t,a} W_1(\phi_E^{t,a,k}(\mathbf{H}_t), \phi_E^{t,k}(\mathbf{H}_t)),$$

where $w_k^{t,a}$ represents the proportion of samples in sub-group $k$ of treatment group $a$, and $\phi_E^{t,k}(\mathbf{H}_t)$ is the uniform mixture of $\{\phi_E^{t,a,k}(\mathbf{H}_t)\}_{a \in \mathcal{A}}$. Note that all the quantities in $L_D^t(\theta_E)$ can be estimated from the observational data. We provide implementation details and our algorithm in Appendix D.

## 6 Experiments

**Main Results.** We evaluate our framework on a fully-synthetic Pharmacokinetic-Pharmacodynamic (PK-PD) benchmark (Bica et al., 2020; Geng et al., 2017) and a semi-synthetic dataset derived from MIMIC-III (Johnson et al., 2016). We integrate our framework into the architectures of the LSTM-based *Counterfactual Recurrent Networks (CRN)* (Bica et al., 2020) and the Transformer-based *Causal Transformer (CT)* (Melnychuk et al., 2022). Across both datasets, our experiments consistently demonstrate that the *synergistic combination* of SGA and RTM achieves state-of-the-art (SOTA) performance in counterfactual outcome estimation (Section 6.1 and 6.2). Full details are deferred to Appendix E.

**Analysis.** We run comprehensive ablation studies, showing **(a)** individual contributions of SGA and RTM (Section 6.3), **(b)** detailed analyses of RTM's masking strategy (Section 6.4), **(c)** RTM's impact on attention mechanisms (Section 6.5), and **(d)** SGA's sensitivity to clustering choices (Section 6.6).

**Baseline Methods.** We compare against SOTA baselines: Marginal Structural Models (**MSMs**) (Hernán et al., 2001; Robins et al., 2000), Recurrent Marginal Structural Networks (**RMSNs**) (Lim, 2018), G-Net (Li et al., 2020), Counterfactual Recurrent Networks (**CRN**) (Bica et al., 2020), and Causal Transformer (**CT**) (Melnychuk et al., 2022).

### 6.1 Experiments on Fully-Synthetic Data from a PK-PD Model of Tumor Growth

We first evaluate on the fully-synthetic data frequently used in the literature (Bica et al., 2020; Melnychuk et al., 2022), which allows simulation of treatment-response dynamics and varying levels of time-dependent confounding.

**Tasks and Evaluation Metrics**. Following Melnychuk et al. (2022), we report normalized Root Mean Squared Error (RMSE) on both *one-step-ahead* and *τ-step-ahead* counterfactual predictions under *varying levels* of time-varying confounding, indexed by $\gamma$. Details on dataset generation, hyperparameters, visualization of the representation space are in Appendix E.1.

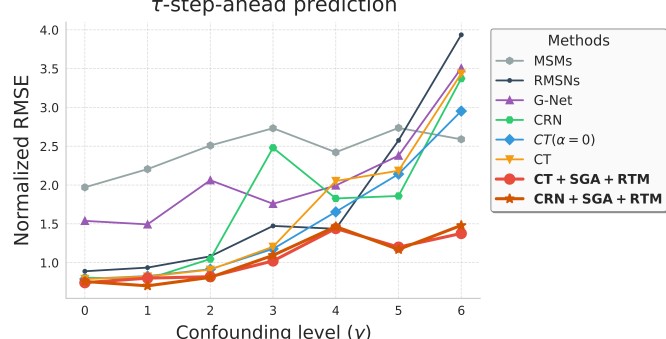

Figure 4: Performances on $\tau$-step-ahead ($\tau$=6) prediction. Note that CT ($\alpha$=0) refers to CT w/o alignment.

**Statistical Significance and Robustness.** To further assess statistical significance and robustness, we additionally report results with error bars in Figure 5.

This figure extends the analysis of Figure 4 by incorporating mean ± standard deviation across multiple runs for the challenging long-term ($\tau = 6$) counterfactual prediction task under varying confounding levels $\gamma$.

**Results.** As shown in Figure 4 and Table 1, integrating both SGA and RTM into CRN and CT **significantly improves** their performance compared to the vanilla models. Notably, our framework performs exceptionally well in scenarios with **high levels of confounding**, indicating our effectiveness in deconfounding. Baselines results for confounding levels $\gamma \in [0, 4]$ are cited from Melnychuk et al. (2022), and we extend experiments to higher confounding levels to more thoroughly test the robustness of our methods in complex causal scenarios. As shown in Figure 5, both CRN+SGA+RTM and CT+SGA+RTM maintain competitive or superior performance with relatively tight error bars as $\gamma$ increases. This demonstrates that our synergistic integration of Sub-treatment Group Alignment (SGA) and Random Temporal Masking (RTM) not only improves predictive accuracy but also enhances stability under increasing time-varying confounding.

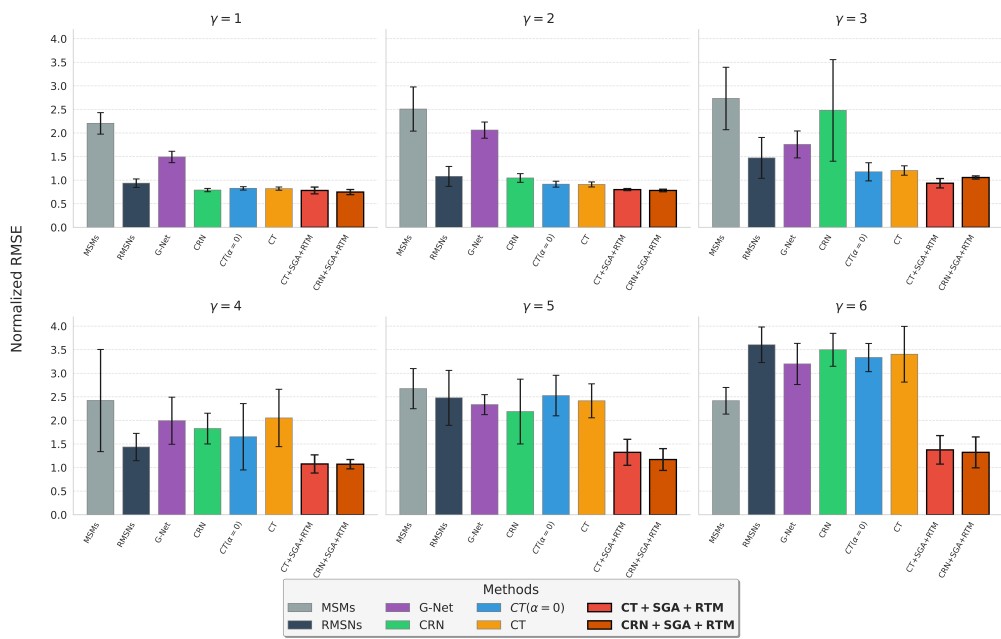

Figure 5: **Impact of varying confounding level $\gamma$ on long-term ($\tau = 6$) prediction performance with error bars**. This figure extends Figure 4 by reporting mean ± standard deviation over multiple runs.

## 6.2 Experiments on Semi-Synthetic Data

We further validate our framework on a semi-synthetic dataset based on real-world medical data from intensive care units. This dataset is generated following the approach of Melnychuk et al. (2022), which builds upon the MIMIC-III dataset (Johnson et al., 2016) to simulate patient trajectories with outcomes that reflect both endogenous and exogenous dependencies while incorporating treatment effects.

**Results and Analysis.** As shown in Table 2, applying our framework on top of CRN and CT **consistently improves performance** compared to the vanilla models. Furthermore, our methods yield performance comparable to the SOTA models. This is consistent with the findings reported in Melnychuk et al. (2022) that confounding may not be the primary challenge in this specific dataset, as there is also minimal difference between the performance of CT and CT($\alpha = 0$). This observation implies that, in this semi-synthetic dataset, the level of confounding may be relatively low. To this end, given that the strength of our approach lies in reducing confounding, it is expected that the performance gain is marginal compared to existing state-of-the-art methods. Details are in Appendix E.2.

Table 1: Normalized RMSE for one-step-ahead and $\tau$-step-ahead predictions on fully synthetic data, comparing vanilla CRN/CT with CRN/CT enhanced with SGA + RTM. The values in **black** indicate lower RMSE for CRN-based models, and values in **violet** indicate lower RMSE for CT-based models.

| $\tau$ | Method | $\gamma = 0$ | $\gamma = 1$ | $\gamma = 2$ | $\gamma = 3$ | $\gamma = 4$ | $\gamma = 5$ | $\gamma = 6$ |
|---|---|---|---|---|---|---|---|---|
| | **CRN** | 0.755 | 0.788 | 0.881 | 1.062 | 1.358 | 1.403 | 2.031 |
| $\tau = 1$ | **CRN + SGA + RTM** | **0.720** | **0.741** | **0.822** | **0.802** | **0.949** | **1.107** | **1.808** |
| | **CT** | 0.770 | 0.783 | 0.864 | 1.098 | 1.413 | 1.497 | 2.204 |
| | **CT + SGA + RTM** | **0.750** | **0.736** | **0.765** | **0.883** | **0.983** | **1.340** | **1.975** |
| | **CRN** | 0.671 | 0.666 | 0.741 | 1.668 | 1.151 | 1.456 | 3.117 |
| $\tau = 2$ | **CRN + SGA + RTM** | **0.629** | **0.648** | **0.674** | **0.822** | **0.934** | **0.948** | **1.308** |
| | **CT** | 0.681 | 0.677 | 0.713 | 0.908 | 1.274 | 1.718 | 3.053 |
| | **CT + SGA + RTM** | **0.610** | **0.673** | **0.688** | **0.751** | **0.786** | **0.911** | **0.983** |
| | **CRN** | 0.700 | 0.692 | 0.818 | 1.959 | 1.360 | 1.684 | 3.451 |
| $\tau = 3$ | **CRN + SGA + RTM** | **0.643** | **0.666** | **0.706** | **0.862** | **0.981** | **0.997** | **1.404** |
| | **CT** | 0.703 | 0.712 | 0.770 | 1.010 | 1.536 | 1.948 | 3.367 |
| | **CT + SGA + RTM** | **0.630** | **0.691** | **0.718** | **0.805** | **0.858** | **1.023** | **1.127** |
| | **CRN** | 0.734 | 0.722 | 0.898 | 2.201 | 1.573 | 1.802 | 3.594 |
| $\tau = 4$ | **CRN + SGA + RTM** | **0.662** | **0.681** | **0.740** | **0.922** | **1.016** | **1.060** | **1.478** |
| | **CT** | 0.726 | 0.748 | 0.822 | 1.089 | 1.762 | 2.095 | 3.516 |
| | **CT + SGA + RTM** | **0.650** | **0.727** | **0.748** | **0.848** | **0.919** | **1.107** | **1.231** |
| | **CRN** | 0.769 | 0.755 | 0.976 | 2.361 | 1.730 | 1.870 | 3.560 |
| $\tau = 5$ | **CRN + SGA + RTM** | **0.682** | **0.700** | **0.771** | **0.983** | **1.046** | **1.118** | **1.497** |
| | **CT** | 0.756 | 0.786 | 0.870 | 1.154 | 1.922 | 2.168 | 3.570 |
| | **CT + SGA + RTM** | **0.677** | **0.761** | **0.779** | **0.889** | **0.995** | **1.162** | **1.315** |
| | **CRN** | 0.807 | 0.790 | 1.047 | 2.480 | 1.827 | 1.859 | 3.372 |
| $\tau = 6$ | **CRN + SGA + RTM** | **0.701** | **0.715** | **0.792** | **1.031** | **1.064** | **1.170** | **1.480** |
| | **CT** | 0.789 | 0.821 | 0.909 | 1.205 | 2.052 | 2.184 | 3.436 |
| | **CT + SGA + RTM** | **0.705** | **0.800** | **0.801** | **0.935** | **1.058** | **1.200** | **1.376** |

Table 2: RMSE for one-step-ahead and $\tau$-step-ahead predictions on semi-synthetic data based on real-world medical data (MIMIC-III).

| | $\tau = 1$ | $\tau = 2$ | $\tau = 3$ | $\tau = 4$ | $\tau = 5$ | $\tau = 6$ | $\tau = 7$ | $\tau = 8$ | $\tau = 9$ | $\tau = 10$ |
|---|---|---|---|---|---|---|---|---|---|---|
| **MSMs** | 0.37 | 0.57 | 0.74 | 0.88 | 1.14 | 1.95 | 3.44 | $> 10.0$ | $> 10.0$ | $> 10.0$ |
| **RMSNs** | 0.24 | 0.47 | 0.60 | 0.70 | 0.78 | 0.84 | 0.89 | 0.94 | 0.97 | 1.00 |
| **G-Net** | 0.34 | 0.67 | 0.83 | 0.94 | 1.03 | 1.10 | 1.16 | 1.21 | 1.25 | 1.29 |
| **CRN** | 0.30 | 0.48 | 0.59 | 0.65 | 0.68 | 0.71 | 0.72 | 0.74 | 0.76 | 0.78 |
| **CRN + SGA + RTM** | **0.27** | **0.43** | **0.52** | **0.58** | **0.62** | **0.65** | **0.67** | **0.69** | **0.72** | **0.73** |
| **CT ($\alpha = 0$)** | **0.20** | **0.38** | 0.46 | **0.50** | **0.52** | 0.54 | 0.56 | **0.57** | 0.59 | 0.60 |
| **CT** | 0.21 | **0.38** | 0.46 | **0.50** | 0.53 | 0.54 | **0.55** | **0.57** | **0.58** | 0.59 |
| **CT + SGA + RTM** | 0.21 | **0.38** | **0.44** | **0.50** | **0.52** | **0.52** | 0.56 | **0.57** | **0.58** | **0.58** |

## 6.3 Ablation Studies

We run ablation experiments to evaluate the individual contributions of **SGA** and **RTM**.

**Effectiveness of SGA.** The left panel of Table 3 demonstrates that incorporating SGA **alone** consistently improves performance over baseline models. The improvements are particularly evident in settings with **higher** confounding levels (larger $\gamma$). This supports our claim that SGA's fine-grained sub-treatment group alignment leads to more effective deconfounding.

**Effectiveness of RTM.** The right panel of Table 3 shows that adding RTM consistently improves model performance. RTM's impact is more pronounced on **longer-term** predictions (larger $\tau$) and under **stronger** confounding (larger $\gamma$), confirming that RTM promotes better temporal generalization.

## 6.4 Analysis of Masking Strategies

To evaluate our choice of Gaussian noise for RTM, we compare it against alternative masking strategies: (*i*) *masking with zeros* and (*ii*) *interpolation-masking* that replaces input covariates from selected time-point $t$ with the **average** covariates of $t - 1$ and $t + 1$. We observe the followings:

**O1. Gaussian noise significantly outperforms other choices.** We hypothesize that this is because Gaussian noise acts as a more effective regularizer. It discourages the model from overfitting to specific input

Table 3: Normalized RMSE for $\tau$-step-ahead predictions on fully-synthetic data, CRN/CT vs. CRN/CT+SGA (left) and CRN/CT vs. CRN/CT+RTM (right).

| $\gamma$ | Method | $\tau = 3$ | $\tau = 4$ | $\tau = 5$ | $\tau = 6$ |
|---|---|---|---|---|---|
| 2 | CRN | 0.818 | 0.898 | 0.976 | 1.047 |
| | CRN + SGA | **0.698** | **0.743** | **0.782** | **0.810** |
| | CT | 0.770 | 0.822 | 0.870 | 0.909 |
| | CT + SGA | **0.762** | **0.813** | **0.854** | **0.876** |
| 6 | CRN | 3.451 | 3.594 | 3.560 | 3.372 |
| | CRN + SGA | **3.157** | **3.284** | **3.231** | **3.034** |
| | CT | 3.367 | 3.516 | 3.570 | 3.436 |
| | CT + SGA | **3.231** | **3.345** | **3.238** | **3.014** |

| $\gamma$ | Method | $\tau = 3$ | $\tau = 4$ | $\tau = 5$ | $\tau = 6$ |
|---|---|---|---|---|---|
| 2 | CRN | 0.818 | 0.898 | 0.976 | 1.047 |
| | CRN + RTM | **0.791** | **0.862** | **0.907** | **0.934** |
| | CT | 0.770 | 0.822 | 0.870 | 0.909 |
| | CT + RTM | **0.720** | **0.752** | **0.787** | **0.819** |
| 6 | CRN | 3.451 | 3.594 | 3.560 | 3.372 |
| | CRN + RTM | **2.316** | **2.469** | **2.541** | **2.523** |
| | CT | 3.367 | 3.516 | 3.570 | 3.436 |
| | CT + RTM | **1.883** | **2.009** | **2.039** | **2.005** |

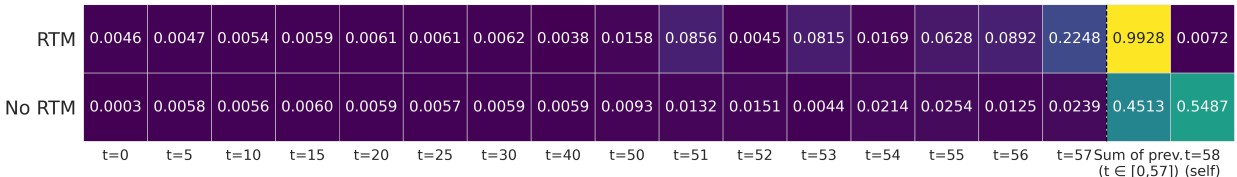

Figure 6: **Heatmap of self-attention weights. Left:** attention to past time points. **Rightmost columns:** sum of total past attention (`Sum of prev.`), and attention to the current time point (`t=58`).

values at the masked time steps and forces it to learn more generalizable representations from the historical context.

Table 4: Comparison of masking strategies on fully-synthetic data ($\gamma$=6) with normalized RMSE. Gaussian noise refers to CT+RTM. Other strategies are also applied to the CT model architecture.

| Masking Strategy | $\tau = 3$ | $\tau = 4$ | $\tau = 5$ | $\tau = 6$ |
|---|---|---|---|---|
| Zero masking | 2.821 | 3.072 | 3.122 | 2.986 |
| Interpolation | 3.435 | 3.556 | 3.602 | 3.480 |
| No Masking | 3.367 | 3.516 | 3.570 | 3.436 |
| Gaussian noise | **1.883** | **2.009** | **2.039** | **2.005** |

**O2. Zero masking performs sub-optimally.** It introduces a strong, and often unnatural signal at zero. Models risk using these artificial zeros or misinterpreting them, which negatively affects the learning of true underlying patterns.

**O3. Interpolation yields the highest error.** It creates artificially smooth data. This obscures natural data variability and can potentially impose a deterministic bias, preventing the model from capturing true temporal dynamics.

### 6.5 RTM Attention Weight Analysis

To investigate how RTM promotes learning of long-range dependencies, we show the softmaxed self-attention weights from the final attention layer of Causal Transformer model. Figure 6 compares attention patterns for CT+RTM versus CT (no RTM) when predicting the outcome at the last time-point ($t = 58$) under high confounding ($\gamma = 6$). This analysis confirms RTM's success in shifting the model's reliance from current time step towards historical context, thereby promoting temporal generalization and the learning of long-term causal patterns, as Figure 6 shows:

- **With RTM, the model allocates over 99% of its attention to previous time-points**. This distributed attention across historical data demonstrates effective leveraging of past information.

- **Without RTM, attention to past steps is significantly lower,** with the model focusing over 54% of its attention on the current time-point. This highlights the tendency of models without RTM to **over-rely on information at the current time step** rather than learn long-term causal patterns.

### 6.6 Clustering Sensitivity

In our experiments, SGA uses Gaussian Mixture Models (GMMs) for *treatment-agnostic* clustering to identify sub-treatment groups. We analyze SGA's sensitivity to the choice of clustering algorithm and the number of clusters (K). Specifically, we compare GMM with k-means using varying K.

Evaluations are performed on the fully-synthetic dataset under the challenging setting of **long-term predictions** ($\tau$=6) and **higher confounding** ($\gamma$=6). We observe from Table 5:

**O1.** SGA's performance **remains relatively stable** across different numbers of clusters, suggesting low sensitivity to this hyperparameter.

**O2. Both k-means and GMM yield comparable performance**, indicating robustness to the specific clustering algorithm used.

Table 5: Sensitivity analysis of SGA to the choice of clustering algorithm and the number of clusters (K). The table reports the normalized RMSE with $\gamma = 6$ at $\tau = 6$.

| Model | Clustering Algorithm | No SGA (K=1) | SGA with varying $K$ | | | |
|---|---|---|---|---|---|---|
| | | | K=2 | K=3 | K=5 | K=10 |
| **CRN+SGA** | **k-means** | 3.372 | 3.568 | 3.301 | **3.259** | 3.504 |
| | **GMM** | 3.372 | **3.034** | 3.258 | 3.276 | 3.595 |
| **CT+SGA** | **k-means** | 3.436 | 3.154 | **3.008** | 3.206 | 3.059 |
| | **GMM** | 3.436 | **3.014** | 3.423 | 3.158 | 3.082 |

These findings confirm SGA's **robustness** to the clustering algorithm and the number of clusters, making it practical for diverse real-world settings where the optimal number of clusters may be unknown or require tuning.

## 7 Conclusion

We address the critical challenge of counterfactual outcome estimation in time series by introducing two novel, synergistic approaches: Sub-treatment Group Alignment (SGA) and Random Temporal Masking (RTM). SGA tackles time-varying confounding at each time point by aligning fine-grained sub-treatment group distributions, leading to tighter counterfactual error bound up to an additive constant term and more effective deconfounding. Complementarily, RTM promotes temporal generalization and robust learning from historical patterns by randomly masking covariates, encouraging the model to preserve underlying causal relationships across time steps and rely less on potentially noisy contemporaneous time steps.

Our comprehensive experiments demonstrate that SGA and RTM are broadly applicable and significantly enhance existing state-of-the-art methods. While each approach individually improves performance, their synergistic combination consistently achieves SOTA performance on both synthetic and semi-synthetic benchmark datasets. Together, SGA and RTM offer a flexible and effective framework for improving causal inference from observational time series data.

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

# A  Potential Outcomes Framework with Time-Varying Treatments and Outcomes

Building on the potential outcomes framework (Rosenbaum & Rubin, 1983; Rubin, 2005), we extend these assumptions to accommodate time-varying treatments and outcomes, following Robins & Hernan (2008).

**Assumption A.1.** (Consistency) If $\bar{\mathbf{A}}_t = \bar{\mathbf{a}}_t$ is a given sequence of treatments for some patient, then $\mathbf{Y}_{t+1}[\bar{\mathbf{a}}_t] = \mathbf{Y}_{t+1}$ This means an individual's potential outcome under the observed treatment history is the observed outcome.

**Assumption A.2.** (Sequential Positivity) Positivity states that there is non-zero probability or not receiving any of the counterfactual treatment. It can be expressed as $0 < P(\mathbf{A}_t = \mathbf{a}_t | \bar{\mathbf{H}}_t = \bar{\mathbf{h}}_t) < 1$, if $P(\bar{\mathbf{H}}_t = \bar{\mathbf{h}}_t) > 0$.

**Assumption A.3.** (Sequential Ignorability) There is no unobserved confounding of treatment at any time and any future outcome. This can be expressed as $\mathbf{A}_t \perp\!\!\!\perp \mathbf{Y}_{t+1}[\mathbf{a}_t] | \bar{\mathbf{H}}_t, \forall\, \mathbf{a}_t \in \mathcal{A}$.

Using assumptions 2.1–2.3, Robins (1986) establishes the sufficient conditions for identifiability through G-computation, ensuring that causal effects can be appropriately identified.

Fig 2 visualizes Causal Directed Acyclic Graphs (DAGs) illustrating causal relationships.

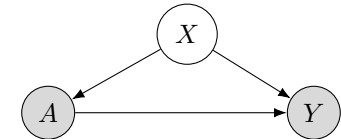

(a) Static (non-time-series) Scenario
*In the static (non-time-series) scenario, we have A as the treatment assignment, X as the covariate, and Y as the outcome.*

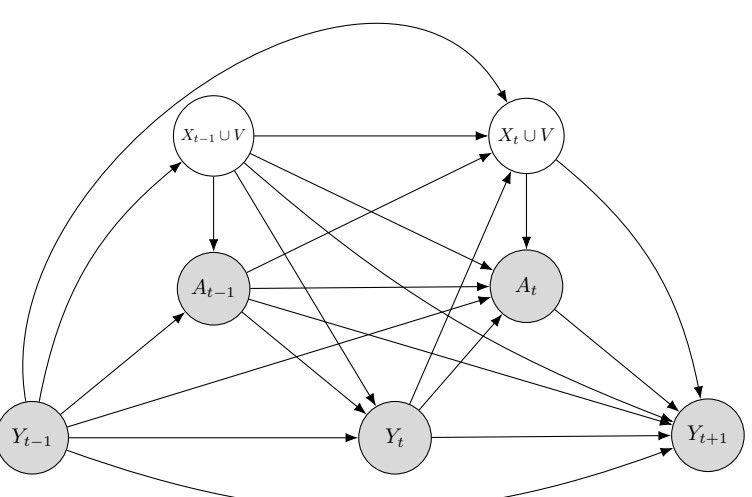

(b) Time-Series Scenario
*In the time-series scenario, $A_t$ is the treatment at time $t$, $X_t \cup V$ represents observed covariates at time $t$, and $Y_t$ is the outcome at time $t$. Here, $V$ denotes static covariates that do not change over time. The diagrams capture the dynamics of treatment effects over time, showing how each component influences subsequent outcomes within the causal framework.*

Figure 7: **Causal Directed Acyclic Graphs (DAGs) Illustrating Causal Relationships. (a)** demonstrate a static (non-time-series) scenario. **(b)** illustrates a time-series scenario.

## B    Related Work

**Estimating counterfactual outcomes under static scenarios.** Many methods have been proposed to learn a *balanced* representation that aligns the distributions across treatment groups, effectively addressing confounding in static settings. A foundational work in this area, CFRNet proposed by Shalit et al. (2017), establishes a counterfactual error bound illustrating that the expected error in estimating individual treatment effects (ITE) is bounded by the sum of its standard generalization error and the discrepancy between treatment group distributions induced by the representation. This concept has been further explored in several subsequent studies on deep causal inference (Yao et al., 2018; Kallus, 2020; Du et al., 2021). However, these methods primarily focus on static data, and their approach of aligning overall treated and control group distributions may not sufficiently adaptable to time-series data (Hernán et al., 2000; Mansournia et al., 2012), where time-dependent confounders make it difficult to disentangle the true effect of a treatment from these caused by the confounding variables.

**Estimating counterfactual outcomes over time.** Estimating counterfactual outcomes in time-series data is challenging due to time-varying confounders. Traditional methods such as G-computation and marginal structural models (Robins, 1986; Robins et al., 2000; Hernán et al., 2001; Robins & Hernan, 2008; Xu et al., 2016) often lack flexibility for complex datasets and rely on strong assumptions. To address these limitations, researchers have developed models that build on the potential outcomes framework initially proposed by Rubin (Rubin, 1978) and extended to time series by Robins & Hernan (2008). Notable among recent methods are Recurrent Marginal Structural Networks (RMSNs) (Lim, 2018), G-Net (Li et al., 2020), Counterfactual Recurrent Networks (CRN) (Bica et al., 2020), and the Causal Transformer (CT) (Melnychuk et al., 2022), which use approaches such as propensity networks and adversarial learning to mitigate the effects of time-varying confounding. The CRN employs recurrent neural networks like LSTMs, while the CT uses Transformer-based architectures, representing the state-of-the-art in this domain. However, practical challenges with adversarial training can affect the stability of causal effect estimations. Specifically, training adversarial networks can be challenging due to issues such as mode collapse and oscillations (Liang et al., 2018). Additionally, adversarial training minimizes the Jensen-Shannon divergence (JSD) only when the discriminator is optimal (Arjovsky & Bottou, 2017), which may not always be achievable in practice; even when the discrminator is optimal, using JSD optimizing relatively loose upper bounds on the counterfactual error. To address these challenges, we propose using the Wasserstein-1 distance. The Wasserstein distance is bounded above by the Kullback-Leibler divergence (JSD is a symmetrized and smoothed version of the Kullback–Leibler divergence) and provides stronger theoretical guarantees (Redko et al., 2017; Mansour et al., 2012). Moreover, the Wasserstein distance has stable gradients even when the compared distributions are far apart (Gulrajani et al., 2017), which enhances training stability and effectiveness.

**Masked language modeling.** Masked language modeling (MLM) is a common self-supervised pre-training technique for large language models. It operates by randomly masking certain words or tokens in the input, with the model trained to predict the masked tokens. BERT (Devlin, 2018) is the most well-known model that employs this technique. Recent studies have also demonstrated the effectiveness of MLM in enhancing generalization across various sequence-based tasks. For example, Chaudhary et al. (2020) showed that when combined with cross-lingual dictionaries, MLM not only improves predictions for the original masked word but also generalizes to its cross-lingual synonyms. Additionally, Czinczoll et al. (2024) illustrated how MLM enhances generalization in long-document tasks by leveraging higher-level semantic representations. Inspired by the success of masking strategies in language models, we introduce Random Temporal Masking (RTM) for time-series data. Unlike MLM, which focuses on predicting the masked inputs, RTM encourages the model to focus on information that is crucial not only for the current time point but also for future time points, preserve causal information, and reduce the risk of overfitting to factual outcomes.

## C    Theorems and Proofs

**Note.** Throughout this section, we follow the notation of Shalit et al. (2017), where $t \in \{0, 1\}$ denotes the treatment assignment.

**Definition C.1** (Definition A4 in Shalit et al. (2017)). Let $\Phi : \mathcal{X} \to \mathcal{R}$ be a representation function. Let $h : \mathcal{R} \times \{0,1\} \to \mathcal{Y}$ be an hypothesis defined over the representation space $\mathcal{R}$. The expected loss for the unit and treatment pair $(x,t)$ is:

$$\ell_{h,\Phi}(x,t) = \int_{\mathcal{Y}} L(Y_t, h(\Phi(x),t))p(Y_t|x)dY_t,$$

where $L(\cdot,\cdot)$ is a loss function, from $\mathcal{Y} \times \mathcal{Y}$ to $\mathbb{R}_+$.

**Definition C.2** (Definition A5 in Shalit et al. (2017)). The expected factual loss and counterfactual losses of $h$ and $\Phi$ are, respectively:

$$\epsilon_F(h,\Phi) = \int_{\mathcal{X}\times\{0,1\}} \ell_{h,\Phi}(x,t)p(x,t)dxdt,$$

$$\epsilon_{CF}(h,\Phi) = \int_{\mathcal{X}\times\{0,1\}} \ell_{h,\Phi}(x,t)p(x,1-t)dxdt,$$

where $p(x,t)$ is distribution over $\mathcal{X} \times \{0,1\}$

**Definition C.3.** For some $K \geq 0$, the set of $K$-Lipschitz functions denotes the set of functions $f$ that verify:

$$\|f(x) - f(x')\| \leq K\|x - x'\|, \ \forall x, x' \in \mathcal{X}$$

Here, we assume that the hypothesis class $\mathbb{H}$ is a subset of $\lambda_H$-Lipschitz functions, where $\lambda_H$ is a positive constant, and we assume that the true labeling functions are $\lambda$-Lipschitz for some positive real number $\lambda$. Moreover, if $f$ is differentiable, a sufficient condition for $f$ to be $K$-Lipschitz is that $\left\|\frac{\partial f}{\partial x}\right\| \leq K, \forall x \in \mathcal{X}$.

**Assumption C.4** (Assumption A2 in Shalit et al. (2017)). There exists a constant $K > 0$ such that for all $x \in \mathcal{X}, t \in \{0,1\}, \|\frac{\partial p(Y_t|x)}{\partial x}\| \leq K$.

**Assumption C.5** (Assumption A3 in Shalit et al. (2017)). The loss function $L$ is differentiable, and there exists a constant $K_L > 0$ such that $\left|\frac{dL(y_1,y_2)}{dy_i}\right| \leq K_L$ for $i = 1,2$. Additionally, there exists a constant $M$ such that for all $y_2 \in \mathcal{Y}, M \geq \int_{\mathcal{Y}} L(y_1,y_2)dy_1$.

**Definition C.6** (Definition A12 in Shalit et al. (2017)). Let $\frac{\partial \Phi(x)}{\partial x}$ be the Jacobian matrix of $\Phi$ at point $x$, i.e., the matrix of the partial derivatives of $\Phi$. Let $\sigma_{\max}(A)$ and $\sigma_{\min}(A)$ denote respectively the largest and smallest singular values of a matrix $A$. Define $\rho(\Phi) = \frac{\sup_{x\in\mathcal{X}}\sigma_{\max}(\frac{\partial\Phi(x)}{\partial x})}{\sigma_{\min}(\frac{\partial\Phi(x)}{\partial x})}$.

**Definition C.7** (Definition A13 in Shalit et al. (2017)). We will call a representation function $\Phi : \mathcal{X} \to \mathcal{R}$ Jacobian-normalized if $\sup_{x\in\mathcal{X}} \sigma_{\max}(\frac{\partial\Phi(x)}{\partial x}) = 1$

*Remark* C.8. Any non-constant representation function $\Phi$ can be Jacobian-normalized by a simple scalar multiplication.

**Lemma C.9** (Lemma A3 in Shalit et al. (2017)). *Let $u = p(t=1)$, then we have,*

$$\epsilon_F(h,\Phi) = u \cdot \epsilon_F^{t=1}(h,\Phi) + (1-u) \cdot \epsilon_F^{t=0}(h,\Phi)$$

$$\epsilon_{CF}(h,\Phi) = (1-u) \cdot \epsilon_{CF}^{t=1}(h,\Phi) + u \cdot \epsilon_{CF}^{t=0}(h,\Phi)$$

**Lemma C.10** (Decomposition over sub-groups). *Assume the representation $\Phi$ induces $K$ treatment–agnostic clusters so that each mixture weight is shared, i.e., $w_k^0 = w_k^1 = w_k$ with $\sum_{k=1}^K w_k = 1$. Define the sub-domain factual and counterfactual losses*

$$\epsilon_F^k(h,\Phi) = \int_{\mathcal{X}\times\{0,1\}} \ell_{h,\Phi}(x,t)\, p_k(x,t)\, dx\, dt,$$

$$\epsilon_{CF}^k(h, \Phi) = \int_{\mathcal{X} \times \{0,1\}} \ell_{h,\Phi}(x,t)\, p_k(x, 1-t)\, dx\, dt,$$

where $p_k(x,t)$ is the conditional distribution of $(X,T)$ for sub-group $k$. Then

$$\epsilon_F(h, \Phi) = \sum_{k=1}^{K} w_k\, \epsilon_F^k(h, \Phi), \qquad \epsilon_{CF}(h, \Phi) = \sum_{k=1}^{K} w_k\, \epsilon_{CF}^k(h, \Phi).$$

Now using the Definition C.11, we rewrite Lemma A8 from Shalit et al. (2017).

**Definition C.11.** Let $u = p(t = 1)$ be the marginal probability of treatment and assume $0 < u < 1$.

$$\epsilon_F^\star(h, \Phi) = (1 - u)\epsilon_F^{t=1}(h, \Phi) + u\epsilon_F^{t=0}(h, \Phi)$$

**Theorem C.12** (Lemma A8 from Shalit et al. (2017)). *Let $u = p(t = 1)$ be the marginal probability of treatment and assume $0 < u < 1$. Let $\Phi : \mathcal{X} \to \mathcal{R}$ be a one-to-one and Jacobian-normalized representation function. Let $K$ be the Lipschitz constant of the functions $p(Y_t|x)$ on $\mathcal{X}$. Let $K_L$ be the Lipschitz constant of the loss function $L$, and $M$ be as in Assumption C.5. Let $h : R \times \{0,1\} \to Y$ be an hypothesis with Lipschitz constant $bK$:*

$$\epsilon_{CF}(h, \Phi) \leq \epsilon_F^\star(h, \Phi) + 2\left(M\rho(\Phi) + b\right) \cdot K \cdot K_L \cdot W_1(p_\Phi^0, p_\Phi^1), \tag{4}$$

*where $B_\Phi = (M\rho(\Phi) + b) \cdot K \cdot K_L$ is a constant and $p_\Phi^a$ is the distribution of the random variable $\Phi(X)$ conditioned on $A = a$, that is, representations for individuals receiving treatment $a \in \{0,1\}$.*

**Theorem C.13** (Simplified Lemma A8 from Shalit et al. (2017)). *Let $\Phi : \mathcal{X} \to \mathcal{R}$ be a one-to-one and Jacobian-normalized representation function. Let $h : R \times \{0,1\} \to Y$ be a hypothesis with Lipschitz constant:*

$$\epsilon_{CF}(h, \Phi) \leq \epsilon_F^\star(h, \Phi) + 2 \cdot B_\Phi \cdot W_1(p_\Phi^0, p_\Phi^1), \tag{5}$$

*where $B_\Phi$ is a constant and $p_\Phi^a$ is the distribution of the random variable $\Phi(X)$ conditioned on $A = a$, that is, representations for individuals receiving treatment $a \in \{0,1\}$.*

**Definition C.14. Wasserstein Distance.** The Kantorovich-Rubenstein dual representation of the Wasserstein-1 distance (Villani, 2009) between two distributions $p_\Phi^0$ and $p_\Phi^1$ is defined as

$$W_1(p_\Phi^0, p_\Phi^1) = \sup_{\|f\|_L \leq 1} \mathbb{E}_{x \sim p_\Phi^0}[f(x)] - \mathbb{E}_{x \sim p_\Phi^1}[f(x)],$$

where the supremum is over the set of 1-Lipschitz functions (all Lipschitz functions $f$ with Lipschitz constant $L \leq 1$. For notational simplicity, we use $D(X_1, X_2)$ to denote a distance between the distributions of any pair of random variables $X_1$ and $X_2$. For instance, $W_1(\Phi(X_0), \Phi(X_1))$ denotes the Wasserstein-1 distance between the distributions of the random variables $\Phi(X_0)$ and $\Phi(X_1)$ for any transformation $\Phi$.

Next, motivated by the generalization bound Liu et al. (2025) from domain adaptation, we show that **SGA optimizes a finer-grained conditional objective that is theoretically tighter than marginal alignment (Theorem 4.1) up to an additive constant term, while empirically yielding a strictly tighter constraint on counterfactual risk in practice.**

**Theorem 4.2 [Complete]** (SGA Improves Generalization Bounds) *Under the following assumptions:*

***A1.*** *For all $k$, the sub-distributions $P_{\Phi,k}^0$ and $P_{\Phi,k}^1$ are Gaussian distributions with means $m_k^0$ and $m_k^1$, and covariances $\Sigma_k^0$ and $\Sigma_k^1$, respectively. The distance between corresponding sub-distributions is less than or equal to the distance between non-corresponding sub-distributions, i.e., $W_1(P_{\Phi,k}^0, P_{\Phi,k}^1) \leq W_1(P_{\Phi,k}^0, P_{\Phi,k'}^1)$ for $k \neq k'$.*

***A2.*** *The sub-treatment group weights are identical across treatment groups, i.e., $w_k^1 = w_k^0$ for all $k \in [K]$. Moreover, there exists a small constant $\epsilon > 0$, such that $\max_{k \in [K]}(tr(\Sigma_k^0)) \leq \epsilon$ and $\max_{k \in [K]}(tr(\Sigma_k^1)) \leq \epsilon$.*

*The following inequalities hold:*

$$\epsilon_{CF}(h, \Phi) \leq \epsilon_F^\star(h, \Phi) + 2B_\Phi\left(\sum_{k=1}^{K} w_k^1 W_1(P_{\Phi,k}^0, P_{\Phi,k}^1)\right)$$

$$\sum_{k=1}^{K} w_k^1 W_1(P_{\Phi,k}^0, P_{\Phi,k}^1) \leq W_1(p_\Phi^0, p_\Phi^1) + \delta_c,$$

*where $B_\Phi$ is the same constant in Theorem 4.1 and $\delta_c$ is $4\sqrt{\epsilon}$.*

*Proof of Theorem 4.2.* Corollary C.19 proves the first statement while Theorem C.20 proves the second. □

**Definition C.15.** (Wasserstein-like distance between Gaussian Mixture Models) Assume that both $X_0$ and $X_1$ are mixtures of $K$ sub-domains. In other words, we have $p_\Phi^0 = \sum_{k=1}^K w_k^0 P_{\Phi,k}^0$ and $p_\Phi^1 = \sum_{k=1}^K w_k^1 P_{\Phi,k}^1$ where for $a \in 0,1$, $w_k^a$ represents the proportion of the $k$-th sub-distribution in treatment group $a$. $P_{\Phi,k}^a$ denotes the distribution of the representations in the $k$-th sub-group under treatment $a$. We define:

$$MW_1(p_\Phi^0, p_\Phi^1) = \min_{w \in \Pi(\mathbf{w^0}, \mathbf{w^1})} \sum_{k=1}^K \sum_{k'=1}^K w_{k,k'} W_1(P_{\Phi,k}^0, P_{\Phi,k'}^1) \tag{6}$$

where $\mathbf{w^0} \doteq [w_1^0, \ldots, w_K^0]$ and $\mathbf{w^1} \doteq [w_1^1, \ldots, w_K^1]$ belong to $\Delta^K$ (the $K-1$ probability simplex). $\Pi(w^0, w^1)$ represents the simplex $\Delta^{K \times K}$ with marginals $\mathbf{w^0}$ and $\mathbf{w^1}$.

**Lemma C.16** (Extension to Lemma 4.1 of Delon & Desolneux (2020)). *Let $\mu_0 = \sum_{k=1}^{K_0} \pi_0^k \mu_0^k$ with $\mu_0^k = \mathcal{N}(m_0^k, \Sigma_0^k)$ and $\mu_1 = \sum_{k=1}^{K_1} \pi_1^k \delta_{m_1^k}$. Let $\tilde{\mu}_0 = \sum_{k=1}^{K_0} \pi_0^k \delta_{m_0^k}$ ($\tilde{\mu}_0$ only retains the means of $\mu_0$). Then,*

$$MW_1(\mu_0, \mu_1) \leq W_1(\tilde{\mu}_0, \mu_1) + \sum_{k=1}^{K_0} \pi_0^k \sqrt{tr\left(\Sigma_0^k\right)}$$

*where $\pi_{\mathbf{0}} \doteq [\pi_0^1, \ldots, \pi_0^k]$ and $\pi_{\mathbf{1}} \doteq [\pi_1^1, \ldots, \pi_1^k]$ belong to $\Delta^K$ (the $K-1$ probability simplex)*

*Proof.*

$$
\begin{aligned}
MW_1(\mu_0, \mu_1) &= \inf_{w \in \Pi(\pi_0, \pi_1)} \sum_{k,l} w_{k,l} W_1(\mu_0^k, \delta_{m_1^l}) \\
&\leq \inf_{w \in \Pi(\pi_{\mathbf{0}}, \pi_{\mathbf{1}})} \sum_{k,l} w_{k,l} W_2(\mu_0^k, \delta_{m_1^l}) \\
&= \inf_{w \in \Pi(\pi_{\mathbf{0}}, \pi_{\mathbf{1}})} \sum_{k,l} w_{k,l} \left[ \sqrt{||m_1^l - m_0^k||^2 + tr\left(\Sigma_0^k\right)} \right] \\
&\leq \inf_{w \in \Pi(\pi_{\mathbf{0}}, \pi_{\mathbf{1}})} \sum_{k,l} w_{k,l} ||m_1^l - m_0^k|| + \sum_k \pi_0^k \sqrt{tr\left(\Sigma_0^k\right)} \\
&\leq W_1(\tilde{\mu}_0, \mu_1) + \sum_{k=1}^{K_0} \pi_0^k \sqrt{tr\left(\Sigma_0^k\right)}
\end{aligned}
\tag{7}
$$

□

*Remark* C.17. We use $\mu_0$, $\mu_1$, and $\tilde{\mu}_0$ to represent a general scenario for measuring the distance between a Gaussian mixture and a mixture of Diract distributions. In the following proofs, we will utilize the defined notation. For instance, $\mu_0$ can be denoted as $p_\Phi^0$, while $\tilde{\mu}_0$ corresponds to $\tilde{p}_\Phi^0$.

For completeness, we recall the following result, Theorem A.10 and Theorem A.11 of Liu et al. (2025), which we rely on in our analysis.

**Theorem C.18** (Theorem A.10 in (Liu et al., 2025)). *Let $p_\Phi^0$ and $p_\Phi^1$ be two Gaussian mixtures with $p_\Phi^0 = \sum_{k=1}^K w_k^0 P_{\Phi,k}^0$ and $p_\Phi^1 = \sum_{k=1}^K w_k^1 P_{\Phi,k}^1$. For all $k$, $P_{\Phi,k}^0$ / $P_{\Phi,k}^1$ are Gaussian distributions with mean $m_k^0$ / $m_k^1$ and covariance $\Sigma_k^0$ / $\Sigma_k^1$. If for $\forall$ $k, k'$, we assume there exists a small constant $\epsilon > 0$, such that $\max_k(trace(\Sigma_k^0)) \leq \epsilon$ and $\max_{k'}(trace(\Sigma_{k'}^1)) \leq \epsilon$. then:*

$$MW_1(p_\Phi^0, p_\Phi^1) \leq W_1(p_\Phi^0, p_\Phi^1) + 4\sqrt{\epsilon} \tag{8}$$

*Proof.* Here, we follow the same structure of the proof for Wassertein-2 in Delon & Desolneux (2020). Let $(P_\phi^0)_n^n$ and $((P_\phi^1)_n^n$ be two sequences of mixtures of Dirac masses respectively converging to $P_\phi^0$ and $P_\phi^1$ in

$\mathcal{P}_1(\mathbb{R}^d)$. Since $MW_1$ is a distance,

$$MW_1(P_\phi^0, P_\phi^1) \leq MW_1((P_\phi^0)^n, (P_\phi^1)^n) + MW_1(P_\phi^0, (P_\phi^0)^n) + MW_1(P_\phi^1, (P_\phi^1)^n)$$
$$= W_1((P_\phi^0)^n, (P_\phi^1)^n) + MW_1(P_\phi^0, (P_\phi^0)^n) + MW_1(P_\phi^1, (P_\phi^1)^n)$$

We can study the limits of these three terms when n $\to +\infty$

First, observe that $MW_1(P_\phi^0, P_\phi^1) = W_1((P_\phi^0)^n, (P_\phi^1)^n) \underset{n\to+\infty}{\to} W_1(P_\phi^0, P_\phi^1)$ since $W_1$ is continuous on $\mathcal{P}_1(\mathbb{R}^d)$.

Second, based on Lemma C.16, we have that

$$MW_1(P_\phi^0, (P_\phi^0)^n) \leq W_1(\tilde{P}_\phi^0, (P_\phi^0)^n) + \sum_{k=1}^K w_k^0 \sqrt{\operatorname{tr}(\Sigma_k^0)} \underset{n\to+\infty}{\to} W_1(\tilde{P}_\phi^0, P_\phi^0) + \sum_{k=1}^K w_k^0 \sqrt{\operatorname{tr}(\Sigma_k^0)}$$

We observe that $x \mapsto \sqrt{x}$ is a concave function, thus by Jensen's inequality, we have that

$$\sum_{k=1}^K w_k^0 \sqrt{\operatorname{tr}(\Sigma_k^0)} \leq \sqrt{\sum_{k=1}^K w_k^0 \operatorname{tr}(\Sigma_k^0)}$$

Also By Jensen's inequality, we have that,

$$W_1(\tilde{P}_\phi^0, P_\phi^0) \leq W_2(\tilde{P}_\phi^0, P_\phi^0).$$

And from Proposition 6 in (Delon & Desolneux, 2020), we have

$$W_2(\tilde{P}_\phi^0, P_\phi^0) \leq \sqrt{\sum_{k=1}^K w_k^0 \operatorname{tr}(\Sigma_k^0)}$$

Similarly for $MW_1(P_\phi^1, (P_\phi^1)^n)$ the same argument holds. Therefore we have,

$$\lim_{n\to\infty} MW_1(P_\phi^0, (P_\phi^0)^n) \leq 2\sqrt{\sum_{k=1}^K w_k^0 \operatorname{tr}(\Sigma_k^0)}$$

And

$$\lim_{n\to\infty} MW_1(P_\phi^1, (P_\phi^1)^n) \leq 2\sqrt{\sum_{k=1}^K w_k^1 \operatorname{tr}(\Sigma_k^1)}$$

We can conclude that:

$$MW_1(P_\phi^0, P_\phi^1) \leq \lim_{n\to\infty} \inf (W_1((P_\phi^0)^n, (P_\phi^1)^n) + MW_1(P_\phi^0, (P_\phi^0)^n) + MW_1(P_\phi^1, (P_\phi^1)^n))$$

$$\leq W_1(P_\phi^0, P_\phi^1) + 2\sqrt{\sum_{k=1}^K w_k^0 \operatorname{tr}(\Sigma_k^0)} + 2\sqrt{\sum_{k=1}^K w_k^1 \operatorname{tr}(\Sigma_k^1)}$$

$$\leq W_1(P_\phi^0, P_\phi^1) + 4\sqrt{\epsilon}$$

This concludes the proof. $\square$

**Corollary C.19** (Sub-domain version of Theorem 4.1). *For every subgroup k the bound of Theorem 4.1 applies:*

$$\epsilon_{CF}^k(h, \Phi) \leq \epsilon_F^{\star\,k}(h, \Phi) + 2B_\Phi W_1(P_{\Phi,k}^0, P_{\Phi,k}^1).$$

*Multiplying by $w_k$ and summing over k, and using Lemma C.10, yields*

$$\epsilon_{CF}(h, \Phi) \leq \epsilon_F^\star(h, \Phi) + 2B_\Phi \left( \sum_{k=1}^K w_k W_1(P_{\Phi,k}^0, P_{\Phi,k}^1) \right)$$

*Remark.* Because the identification of the sub-treatment group is *treatment-agnostic*, each unit's subgroup index is determined *solely* by the representation $\Phi(x)$, independent of the treatment label $A$. Consequently, whenever treatment assignment is randomized or conditionally ignorable given the representation ($A \perp k \mid \Phi(x)$), the probability of falling into cluster $k$ is identical across arms: $\Pr(k \mid A{=}0) = \Pr(k \mid A{=}1) = w_k$, so the use of $w_k$, $w_k^0$ and $w_k^1$ is interchangeable.

**Theorem C.20** (SGA Improves Generalization Bounds, Theorem A.11 in (Liu et al., 2025))**.** *Under the following assumptions:* ***A1.*** *For all $k$, the sub-distributions $P_{\Phi,k}^0$ and $P_{\Phi,k}^1$ are Gaussian distributions with means $m_k^0$ and $m_k^1$, and covariances $\Sigma_k^0$ and $\Sigma_k^1$, respectively. The distance between corresponding sub-distributions is less than or equal to the distance between non-corresponding sub-distributions, i.e., $W_1(P_{\Phi,k}^0, P_{\Phi,k}^1) \leq W_1(P_{\Phi,k}^0, P_{\Phi,k'}^1)$ for $k \neq k'$.*

***A2.*** *There exists a small constant $\epsilon > 0$, such that $\max_{1 \leq k \leq K}(tr(\Sigma_k^0)) \leq \epsilon$ and $\max_{1 \leq k \leq K}(tr(\Sigma_k^1)) \leq \epsilon$. Then the following inequalities hold:*

$$\sum_{k=1}^K w_k^1 W_1(P_{\Phi,k}^0, P_{\Phi,k}^1) \leq W_1(p_\Phi^0, p_\Phi^1) + \delta_c,$$

*where $\delta_c$ is $4\sqrt{\epsilon}$.*

*Proof.* Let $\mathbf{w^0} \doteq [w_1^0, \ldots, w_K^0]$ and $\mathbf{w^1} \doteq [w_1^1, \ldots, w_K^1]$ belong to $\Delta^K$ (the $K-1$ probability simplex). $\Pi(w^0, w^1)$ represents the simplex $\Delta^{K \times K}$ with marginals $\mathbf{w^0}$ and $\mathbf{w^1}$. For any $w \in \Pi(w^0, w^1)$, we can express $w_k^1 = \sum_{k'=1}^K w_{k,k'}$. Based on assumption A1, we have:

$$\sum_{k=1}^K w_k^1 W_1(P_{\Phi,k}^0, P_{\Phi,k}^1) = \sum_{k=1}^K \sum_{k'=1}^K w_{k,k'} W_1(P_{\Phi,k}^0, P_{\Phi,k}^1)$$
$$\leq \sum_{k=1}^K \sum_{k'=1}^K w_{k,k'} W_1(P_{\Phi,k}^0, P_{\Phi,k'}^1).$$

Thus, we have (with $MW_1(p_\Phi^0, p_\Phi^1)$ defined in Appendix C.15):

$$\sum_{k=1}^K w_k^1 W_1(P_{\Phi,k}^0, P_{\Phi,k}^1) \leq \min_{w \in \Pi(\mathbf{w^0}, \mathbf{w^1})} \sum_{k=1}^K \sum_{k'=1}^K w_{k,k'} W_1(P_{\Phi,k}^0, P_{\Phi,k'}^1) \tag{9}$$
$$= MW_1(p_\Phi^0, p_\Phi^1).$$

From Theorem C.18, we have:

$$MW_1(p_\Phi^0, p_\Phi^1) \leq W_1(p_\Phi^0, p_\Phi^1) + 4\sqrt{\epsilon}. \tag{10}$$

Combining the above results:

$$\sum_{k=1}^K w_k^1 W_1(P_{\Phi,k}^0, P_{\Phi,k}^1) \leq W_1(p_\Phi^0, p_\Phi^1) + 4\sqrt{\epsilon}. \tag{11}$$

$\square$

**Justifications of Assumptions in Theorem 4.2.** Theorem 4.2 uses the assumptions regarding the distributional properties of sub-treatment groups in the learned representation space $\mathcal{R}$. Specifically, Assumption **A1** posits that sub-distributions $P_{\Phi,k}^0$ and $P_{\Phi,k}^1$ (for treatment arms 0, 1) are Gaussian and that $W_1(P_{\Phi,k}^0, P_{\Phi,k}^1) \leq W_1(P_{\Phi,k}^0, P_{\Phi,k'}^1)$ for $k \neq k'$. Assumption **A2** states that the trace of their covariances is bounded by a small $\epsilon$. We justify the validity of these assumptions below:

1. **Gaussian Sub-distributions in Learned Representation Space (A1, first part):** The assumption that sub-treatment groups exhibit Gaussian distributions is made primarily for mathematical tractability in proving the tighter bound. While raw input covariates in complex real-world or semi-synthetic datasets are unlikely to be perfectly Gaussian, this assumption applies to the distributions *after* they have been processed by the encoder network $\phi_E$ into the representation space $\Phi(\mathbf{H}_t)$.

   - **Effect of Deep Encoders:** Deep neural networks, through multiple layers of non-linear transformations, can often map complex, high-dimensional input distributions into lower-dimensional latent spaces where the resulting distributions of distinct sub-group tend to be more regular and sometimes approximate unimodal, bell-shaped forms.
   - **Support from Latent Representations Figures:** The histograms in Figure 8 show the learned representations for (treatment, cluster) combinations at selected timepoints. Here we use PCA to map the latent representations to low dimensions. Many of these distributions (Figure 8), exhibit clear unimodal and symmetric shapes, strongly suggestive of an underlying Gaussian or near-Gaussian structure in that principal component's direction.
   - **Robustness of SGA Implementation:** Importantly, the practical implementation of SGA does not strictly require perfect Gaussianity. Our sensitivity analysis in Section 6.6 shows that SGA performs well when using k-means for clustering, which imposes fewer explicit distributional assumptions than GMM. This empirical robustness suggests that the benefits of SGA extend beyond scenarios where the Gaussian assumption holds perfectly.

2. **Distance Assumption (A1, second part):** The assumption $W_1(P_{\Phi,k}^0, P_{\Phi,k}^1) \leq W_1(P_{\Phi,k}^0, P_{\Phi,k'}^1)$ for $k \neq k'$ (where $0, 1$ are distinct treatment arms) posits that the Wasserstein-1 distance between *corresponding* sub-groups (same $k$) across two different treatment arms is less than or equal to the distance between a sub-group in one arm and a *non-corresponding* sub-group (different $k'$) in the other arm.

   We provide strong empirical evidence supporting this assumption from our fully-synthetic dataset experiments. For each combination of timepoint, sub-group $k$, and pair of distinct treatment arms $(A, B)$, we calculated the $W_1(P_{\Phi,k}^A, P_{\Phi,k}^B)$ (paired distance) and the average $W_1(P_{\Phi,k}^A, P_{\Phi,k' \neq k}^B)$ (average non-paired distance). As shown in Table 6, the paired Wasserstein distance is less than the average non-Paired Wasserstein distance. Table 6 presents a detailed view of all such pairwise comparisons for selected representative timepoints (early, mid, and late). This strongly supports Assumption A1 by demonstrating that corresponding sub-groups are indeed closer to each other than to non-corresponding ones in the learned representation space.

3. **Bounded Covariance Trace (A2):** This assumption $\left( \max_{1 \leq k \leq K} (\mathrm{tr}(\Sigma_k^a)) \leq \epsilon \text{ for any treatment arm } a \right)$ implies that the learned representations of sub-treatment groups are relatively compact. This is a desirable property of representations since well-learned representations for distinct sub-groups are expected to be somewhat concentrated.

## D   Implementation Details and Algorithm

**Training Procedure with Warmup and Epoch-wise SGA**   In order to stabilize the encoder representations before introducing the Sub-treatment Group Alignment (SGA) objective, we use a *warmup* phase where we optimize a baseline method's losses (e.g., factual outcome prediction plus any other baseline objectives) *without* the SGA term. This is because early in training, the representations encoded by $\theta_E$ may be very noisy and prone to random alignment. This ensures that the model achieves a coherent representation of the data before the sub-treatment groups are forced to align. Once these representations are sufficiently stable, we then *periodically* compute and backpropagate SGA at specific epochs (e.g., every gap_epoch). This strategy limits the noise in cluster assignments and reduces the computational overhead of repeatedly clustering on each iteration.

**Computing the Uniform Mixture of Sub-treatment Groups**   In our implementation of the SGA loss, for each time step $t$ and each cluster $k$, we compute the uniform mixture of sub-treatment groups $\phi_E^{t,k}$.

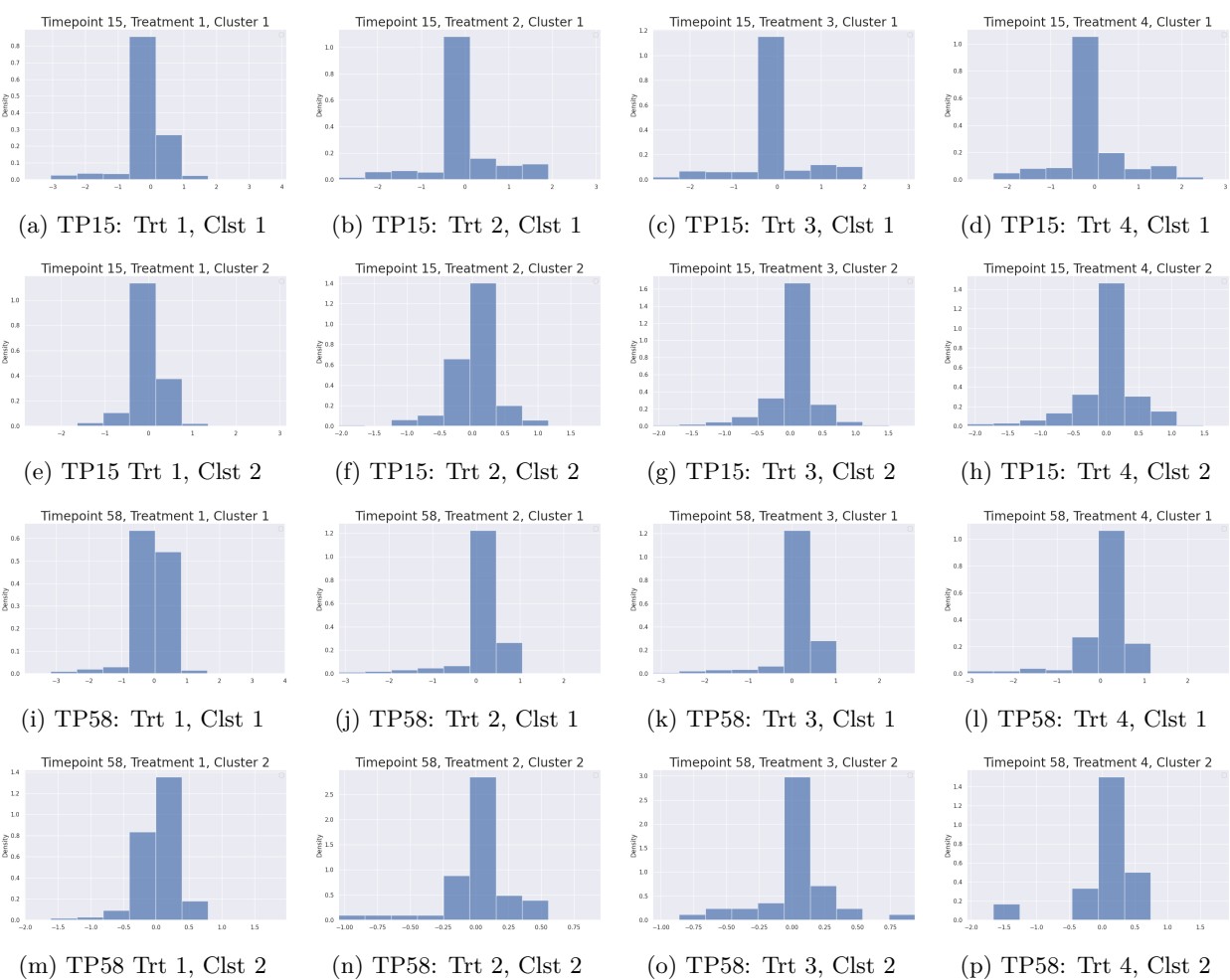

Figure 8: **Distributions of learned representations** for sub-groups within each treatment arm.

Table 6: **Detailed empirical validation of the distance assumption** $(W_1(P^A_{\Phi,k}, P^B_{\Phi,k}) \leq W_1(P^A_{\Phi,k}, P^B_{\Phi,k'})$ for $k \neq k')$ from the fully-synthetic dataset on high confounding $\gamma=6$. The table shows all pairwise treatment comparisons for selected timepoints and sub-groups.

| Timepoint | Sub-group $k$ | Treat. Pair (A, B) | Paired $W_1$ $(\Phi^A_k, \Phi^B_k)$ | Avg. Non-Paired $W_1$ $(\Phi^A_k, \Phi^B_{k' \neq k})$ | Paired < Non-Paired |
|---|---|---|---|---|---|
| *Timepoint 3* | | | | | |
| 3 | 1 | (1, 2) | 0.041 | 4.744 | True |
| 3 | 1 | (1, 3) | 0.058 | 4.842 | True |
| 3 | 1 | (1, 4) | 0.331 | 6.317 | True |
| 3 | 1 | (2, 3) | 0.024 | 4.672 | True |
| 3 | 1 | (2, 4) | 0.352 | 6.156 | True |
| 3 | 1 | (3, 4) | 0.314 | 6.165 | True |
| 3 | 2 | (1, 2) | 0.567 | 3.563 | True |
| 3 | 2 | (1, 3) | 0.657 | 3.575 | True |
| 3 | 2 | (1, 4) | 2.509 | 4.015 | True |
| 3 | 2 | (2, 3) | 0.119 | 4.585 | True |
| 3 | 2 | (2, 4) | 1.183 | 5.009 | True |
| 3 | 2 | (3, 4) | 1.046 | 5.105 | True |
| *Timepoint 30* | | | | | |
| 30 | 1 | (1, 2) | 0.092 | 3.565 | True |
| 30 | 1 | (1, 3) | 0.099 | 3.458 | True |
| 30 | 1 | (1, 4) | 0.448 | 3.466 | True |
| 30 | 1 | (2, 3) | 0.011 | 3.124 | True |
| 30 | 1 | (2, 4) | 0.178 | 3.132 | True |
| 30 | 1 | (3, 4) | 0.170 | 3.119 | True |
| 30 | 2 | (1, 2) | 0.085 | 3.260 | True |
| 30 | 2 | (1, 3) | 0.126 | 3.246 | True |
| 30 | 2 | (1, 4) | 0.189 | 2.785 | True |
| 30 | 2 | (2, 3) | 0.108 | 3.218 | True |
| 30 | 2 | (2, 4) | 0.168 | 2.756 | True |
| 30 | 2 | (3, 4) | 0.176 | 2.649 | True |
| *Timepoint 59* | | | | | |
| 59 | 1 | (1, 2) | 0.048 | 3.821 | True |
| 59 | 1 | (1, 3) | 0.056 | 3.514 | True |
| 59 | 1 | (1, 4) | 0.288 | 3.306 | True |
| 59 | 1 | (2, 3) | 0.012 | 3.339 | True |
| 59 | 1 | (2, 4) | 0.164 | 3.130 | True |
| 59 | 1 | (3, 4) | 0.138 | 3.091 | True |
| 59 | 2 | (1, 2) | 0.170 | 3.447 | True |
| 59 | 2 | (1, 3) | 0.113 | 3.407 | True |
| 59 | 2 | (1, 4) | 0.310 | 3.037 | True |
| 59 | 2 | (2, 3) | 0.233 | 3.608 | True |
| 59 | 2 | (2, 4) | 0.531 | 3.238 | True |
| 59 | 2 | (3, 4) | 0.269 | 2.929 | True |

To compute this uniform mixture, we perform the following steps:

1. **Concatenate representations across treatments:**
   For the $k$-th cluster at time $t$, we collect the representations from all treatment groups:

   $$\phi_E^{t,k}(\mathbf{H}_t) = \bigcup_{a \in \mathcal{A}} \phi_E^{t,a,k}(\mathbf{H}_t),$$

   where $\phi_E^{t,a,k}(\mathbf{H}_t)$ denotes the representations of samples in the $k$-th sub-group of treatment $a$ at time $t$.

2. **Shuffle and subsample:**
   We shuffle the concatenated representations to ensure that samples from different treatments are thoroughly mixed. Then we select $\frac{1}{|\mathcal{A}|}$ from the concatenated representations as $\phi_E^{t,k}$.

Full details are included in Algorithm 1

**Clustering Correspondence & Clustering Details.** On a high level, we perform a *treatment-agnostic clustering* in the using the learned representations at each time step. In particular, the clustering is done iteratively and end-to-end together with model (network) training.

*A Running Example of Clustering Correspondence.* Specifically, at each time step, we run the clustering algorithm (GMM/K-Means) on all the individuals to partition them into $K$ subgroups. Note that here in each subgroup, we may have individuals within different treatment groups because the clustering is treatment agnostic. For example, suppose there are 20 individuals in treatment group 0 and 30 individuals in treatment group 1. In this case, we run the clustering algorithm on all the 50 (20+30) individuals and partition them in K=3 subgroups, with 15,17,18 individuals in each subgroup respectively, The 15 individuals in the first subgroup may be from different treatment groups (e.g., 7 from treatment group 0 and 8 from treatment group 1).

In this case, these 7 individuals from treatment group 0 and 8 individuals from treatment group 1 are *corresponding sub-treatment groups*, and we have established a *cluster correspondence* between them, as they all belong to the same subgroup identified by the clustering algorithm.

**Hyper-parameters.** Regarding the hyperparameters of RTM, we perform hyperparameter optimization for all benchmarks via random grid search with respect to the factual RMSE of the validation set. For reproducibility,we fixed the masking prob as 5% for all experiments and report the selected hyperparameters in Section E.

Regarding the deep sequence model hyperparameters, to ensure a fair comparison, *we did not tune or re-implement the CT/CRN baseline.* Instead, we directly used the architecture / model hyperparameters reported in the original papers and their official code repositories.

## E    Experiments

### E.1    Fully Synthetic Dataset

#### E.1.1    Dataset Generation

Dataset generation follows the identical setup as Bica et al. (2020); Melnychuk et al. (2022). The tumor growth simulator (Geng et al., 2017) models the tumor volume $Y_{t+1}$ after $t + 1$ days of diagnosis. It includes two binary treatments: (i) radiotherapy $A_t^r$ and (ii) chemotherapy $A_t^c$. These treatments influence tumor progression as follows:

- **Radiotherapy** has an immediate impact, denoted by $d(t)$, on the tumor volume at the next time step.

- **Chemotherapy** impacts future tumor progression with an exponentially decaying effect $C(t)$.

The model is described by the equation:

$$Y_{t+1} = \left( 1 + \rho \log \left( \frac{K}{Y_t} \right) - \beta_c C_t - (\alpha_r d_t + \beta_r d_t^2) + \varepsilon_t \right) Y_t$$

where $\varepsilon_t \sim N(0, 0.01^2)$ is independent noise, and the variables $\beta_c, \alpha_r, \beta_r$ represent the response characteristics for each individual. These parameters are drawn from truncated normal distributions comprising three mixture components. For a full list of parameter values, the code implementation should be consulted.

Time-varying confounding is accounted for through biased treatment assignments, where treatment allocation is identical across both therapies $A_t^r$ and $A_t^c$:

$$A_t^r, A_t^c \sim \text{Bernoulli} \left( \sigma \left( \frac{\gamma}{D_{\max}} (\bar{D}_{15} \bar{Y}_{t-1} - \frac{D_{\max}}{2}) \right) \right)$$

In this formula, $\sigma(\cdot)$ represents the sigmoid function, $D_{\max}$ is the maximum tumor diameter in the last 15 days, and $\gamma$ is the confounding parameter. $\bar{D}_{15}(\bar{Y}_{t-1})$ refers to the average tumor diameter over the previous 15 days. If $\gamma = 0$, the treatment assignment is fully randomized, but for increasing values of $\gamma$, time-varying confounding gradually intensifies. More details can be found in Appendix J in CT Melnychuk et al. (2022).

### E.1.2   Experiments Setup

**One-step-ahead prediction.** To evaluate one-step-ahead predictions, we utilize the counterfactual trajectories simulated in CT. Our approach involves comparing our estimated outcomes $Y_{t+1}$ against all four possible combinations of one-step-ahead counterfactual outcomes. This effectively captures the tumor volumes under every possible treatment assignment at the next time step.

**$\tau$-step-ahead prediction.** For multi-step-ahead predictions, the number of potential outcomes for $Y_{t+2},...,Y_{t+\tau_{max}}$ grows exponentially with the prediction horizon $\tau_{max}$. To manage this complexity, and following the methodology in CT, we employ a single sliding treatment strategy. This approach is motivated by the importance of treatment timing in clinical settings. As discussed in the introduction, consider the treatment of *Ductal Carcinoma In Situ*, where the timing of surgical intervention is critical: delaying surgery might allow the cancer to progress to an invasive stage, while performing it too early could lead to unnecessary invasiveness. To assess whether our models can identify the optimal timing for treatment, we simulate trajectories with a single treatment event that is iteratively shifted across a window ranging from time t to $t + \tau_{max} - 1$.

**Performance evaluation.** In line with Melnychuk et al. (2022), we evaluate model performance using the mean Root Mean Square Error (RMSE) on the test set, which consists of hold-out data. The RMSE is normalized by dividing by the maximum tumor volume $V_{max} = 1150\text{cm}^3$. Additionally, we report the test RMSE calculated exclusively on the counterfactual outcomes following the rolling origin, thereby isolating the evaluation from historical factual patient trajectories.

### E.1.3   Empirical Analysis of our Proposed Generalization Bound

As shown in Fig 9, here we empirically evaluate the proposed generalization bound. we provide empirical evidence that Sub-treatment Group Alignment (SGA) results in a much tighter upper bound compared to the original method in Theorem 4.1.

### E.1.4   Analysis of Representation Space

We visualize the feature spaces learned by our Sub-treatment Group Alignment (SGA) method. As shown in Figure 10, SGA is able to learn treatment-invariant representations, which improves performance in counterfactual outcome estimation.

---

**Algorithm 1** Counterfactual Outcome Estimation with Sub-treatment Group Alignment (SGA) and Random Temporal Masking (RTM)

---

**Require:**

1: $\mathcal{D} = \{(\mathbf{X}_i^t, \mathbf{A}_i^t, \mathbf{Y}_i^{t+1})\}_{i=1}^N$: Training data for $N$ individuals for t = 1,...,T
2: $\theta_E, \theta_Y$: Parameters of encoder $\phi_E$ and regressor $f_Y$
3: $\lambda$: Hyperparameter for $L_D$
4: $K$: Number of sub-treatment groups (clusters)
5: $\mathcal{A}$: Set of possible treatments
6: MaskProb: Probability of masking covariates in RTM
7: $\eta$: Learning rate
8: $\ell(\cdot, \cdot)$: Loss function (e.g., mean squared error)
9: pretrain_epochs: number of epochs to train only $L_Y$ (SGA off)
10: gap_epoch: interval at which SGA is turned on (e.g. 5,10,15,...)
11: max_epochs: total number of training epochs
12: **Apply Random Temporal Masking (RTM):**
13: **for** each time step $t = 1$ to $T$ **do**
14:     With probability MaskProb, replace $X^t$ with Gaussian noise
15: **end for**
16: **for** epoch $= 1$ to max_epochs **do**
17:     Initialize $L_Y = 0, L_D = 0$
18:     **for** each time step $t = 1$ to $T$ **do**
19:         $\Phi_E(\mathbf{H}_t) = \phi_E(\mathbf{H}_t, A^t)$
20:         $\hat{Y} = f_Y(\Phi_E(\mathbf{H}_t))$
21:         **Compute Factual Outcome Loss:**
22:         $L_Y = L_Y + \ell(Y^{t+1}, \hat{Y}^{t+1})$
23:         **if** (epoch $\geq$ pretrain_epochs) **AND** ((epoch mod gap_epoch) $= 0$) **then**
24:             **Compute SGA Loss:**
25:             **for** each treatment $a \in \mathcal{A}$ **do**
26:                 **Cluster representations into $K$ sub-groups:**
27:                 Apply GMM to $\Phi_E(\mathbf{H}_t)$ to obtain clusters $\{\phi_E^{t,a,k}\}_{k=1}^K$
28:                 Compute weights $w_k^{t,a} = \frac{n_k^{t,a}}{n^{t,a}}$, where $n_k^{t,a}$ is the number of samples in cluster $k$, $n^{t,a}$ is the total number of samples with treatment $a$ at time $t$
29:             **end for**
30:             **Compute uniform mixture of sub-groups $\phi_E^{t,k}$**
31:             **Compute SGA loss at time $t$:**
32:             $L_D = L_D + \sum_{k=1}^K \sum_{a \in \mathcal{A}} w_k^{t,a} \cdot W_1\left(\phi_E^{t,a,k}, \phi_E^{t,k}\right)$
33:         **end if**
34:     **end for**
35:     **Compute Total Loss:**
36:     **if** (epoch $\geq$ pretrain_epochs) **AND** ((epoch mod gap_epoch) $= 0$) **then**
37:         $L = L_Y + \lambda L_D$
38:     **else**
39:         $L = L_Y +$ baseline suggested loss
40:     **end if**
41:     $\theta_E \leftarrow \theta_E - \eta \nabla_{\theta_E} L$
42:     $\theta_Y \leftarrow \theta_Y - \eta \nabla_{\theta_Y} L$
43: **end for**

---

### E.1.5 Analysis of Masking Frequency

Here, we highlight the potential impact of RTM masking frequency. Indeed, RTM works on a trade-off: by masking covariates, it forces the model to rely more on historical context, aiming to learn more robust

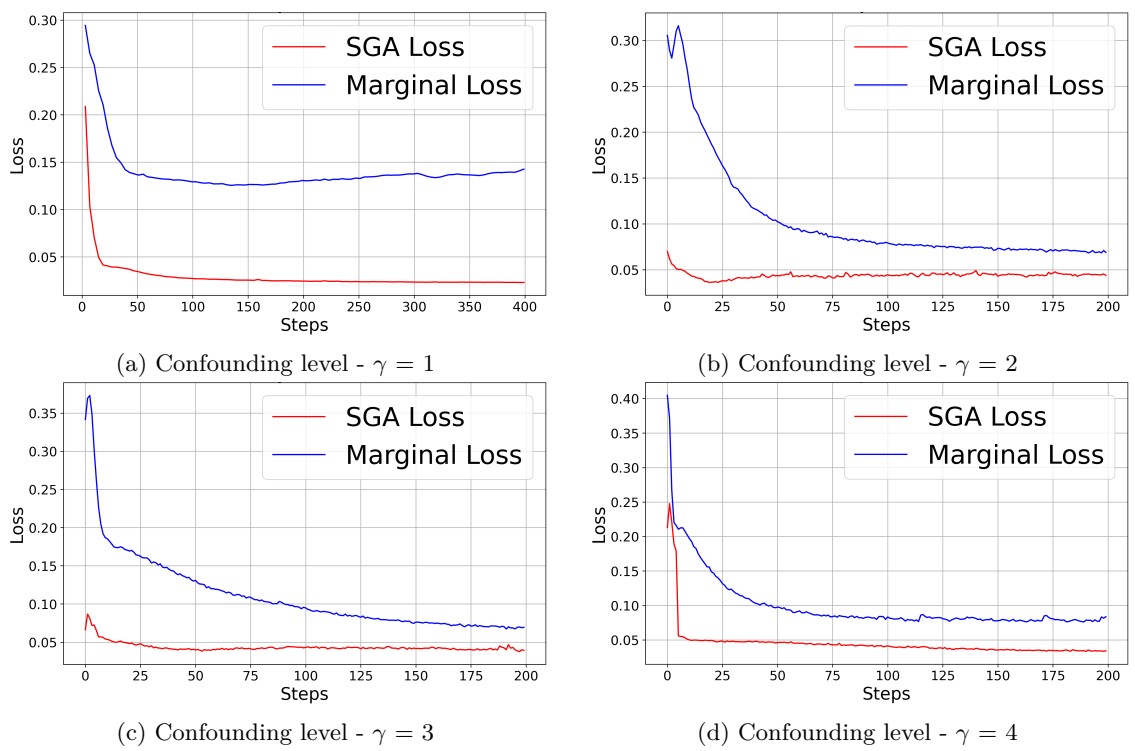

(a) Confounding level - $\gamma = 1$        (b) Confounding level - $\gamma = 2$

(c) Confounding level - $\gamma = 3$        (d) Confounding level - $\gamma = 4$

Figure 9: Empirical results for Sub-treatment Group Alignment (SGA) vs. the original method in Theorem 4.1 with varying confounding levels.

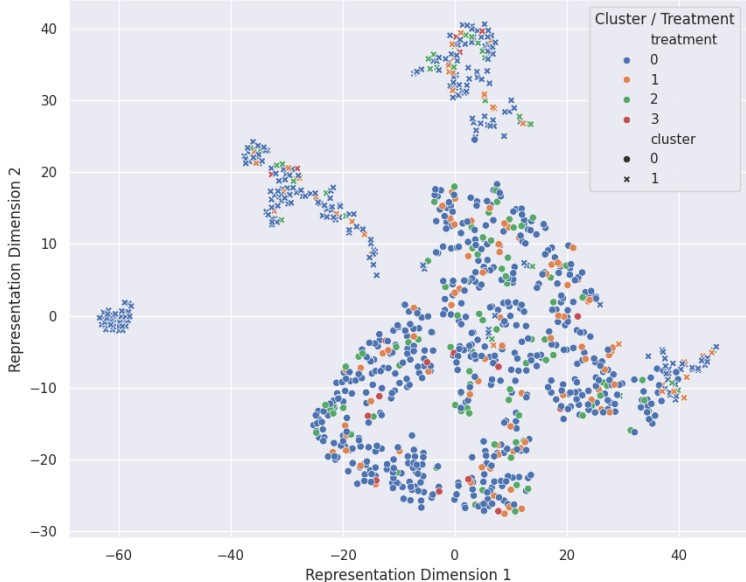

Figure 10: **Representations at the last time point in training under high-confounding scenarios**, with features projected to two dimensions using UMAP.

causal patterns and reduce overfitting to potentially noisy current-step information. However, if this masking becomes too aggressive, the loss of immediate information could potentially degrade performance on tasks heavily reliant on that information. Experiments with varying masking frequencies with high confounding level $\gamma = 6$ on fully synthetic data are shown in Table 7.

Table 7: Performance metrics for **different masking frequencies**.

| Masking Freq | $\tau = 4$ | $\tau = 5$ | $\tau = 6$ |
|---|---|---|---|
| 0% | 3.516 | 3.570 | 3.436 |
| 2% | 2.677 | 2.755 | 2.756 |
| 5% | 2.009 | 2.039 | 2.005 |
| 10% | 2.897 | 2.986 | 3.029 |
| 20% | 3.519 | 3.612 | 3.610 |
| 50% | 6.630 | 6.863 | 7.006 |

As observed, performance initially improves with masking (peaking around 5%) but degrades significantly with higher masking frequencies, confirming the trade-off.

### E.1.6   Error Bars

**Extended Discussion on Sensitivity to varying confounding levels for Long-Term Predictions ($\tau = 6$)**   Figure 11 shows the robustness of our proposed framework when subjected to varying confounding levels, specifically for the challenging task of long-term counterfactual outcome prediction at $\tau = 6$. The Normalized RMSE, along with error bars indicating the spread of results (e.g., mean $\pm$ standard deviation), is reported for our methods (CT+SGA+RTM and CRN+SGA+RTM) and several baselines across six different settings of $\gamma$. The results highlight that both CRN+SGA+RTM and CT+SGA+RTM maintain competitive or superior performance with relatively tight error bars as $\gamma$ changes. This suggests that our synergistic combination of Sub-treatment Group Alignment (SGA) and Random Temporal Masking (RTM) not only achieves high accuracy but also exhibits stability in its predictions for long-range forecasting. The inclusion of error bars provides further confidence in these findings.

### E.1.7   Model Hyperparameters

Benchmark method hyperparameters and performance are sourced from the GitHub repository of Melnychuk et al. (2022).

Table 8: Model hyperparameters used for the fully-synthetic dataset

|  | CT + SGA + RTM | CRN + SGA + RTM |
|---|---|---|
| $\gamma = 0$ | batch size = 2048, lr = 0.025, $\lambda$ = 0.0001, dropout rate = 0.2, Adam | encoder batch size = 1024, encoder lr = 0.005, encoder dropout rate = 0.1, decoder batch size = 4096, decoder lr = 0.01, decoder dropout rate = 0.2, $\lambda$ = 0.0001, Adam |
| $\gamma = 1$ | batch size = 1024, lr = 0.02, $\lambda$ = 0.0001, dropout rate = 0.1, Adam | encoder batch size = 1024, encoder lr = 0.005, encoder dropout rate = 0.1, decoder batch size = 4096, decoder lr = 0.01, decoder dropout rate = 0.1, $\lambda$ = 0.0001, Adam |
| $\gamma = 2$ | batch size = 512, lr = 0.02, $\lambda$ = 0.001, dropout rate = 0.1, Adam | encoder batch size = 1024, encoder lr = 0.005, encoder dropout rate = 0.2, decoder batch size = 4096, decoder lr = 0.01, decoder dropout rate = 0.1, $\lambda$ = 0.0001, Adam |
| $\gamma = 3$ | batch size = 512, lr = 0.03, $\lambda$ = 0.001, dropout rate = 0.1, Adam | encoder batch size = 1024, encoder lr = 0.005, encoder dropout rate = 0.2, decoder batch size = 4096, decoder lr = 0.01, decoder dropout rate = 0.1 , $\lambda$ = 0.001, Adam |
| $\gamma = 4$ | batch size = 1024, lr = 0.01, $\lambda$ = 0.001, dropout rate = 0.1, Adam | encoder batch size = 1024, encoder lr = 0.005, encoder dropout rate = 0.2, decoder batch size = 4096, decoder lr = 0.01, decoder dropout rate = 0.1, $\lambda$ = 0.01, Adam |
| $\gamma = 5$ | batch size = 256, lr = 0.01, $\lambda$ = 0.1, dropout rate = 0.1, Adam | encoder batch size = 64, encoder lr = 0.001, encoder dropout rate = 0.2, decoder batch size = 1024, decoder lr = 0.001, decoder dropout rate = 0.1, $\lambda$ = 0.001, Adam |
| $\gamma = 6$ | batch size = 256, lr = 0.005, $\lambda$ = 0.1, dropout rate = 0.1, Adam | encoder batch size = 64, encoder lr = 0.001, encoder dropout rate = 0.2, decoder batch size = 1024, decoder lr = 0.001, decoder dropout rate = 0.1, $\lambda$ = 0.01, Adam |

τ-step-ahead Prediction

Figure 11: **Impact of varying confounding level $\gamma$ on long-term ($\tau = 6$) prediction performance with error bars**. This figure extends analysis from Figure4 by incorporating error bars.

### E.2 Semi-synthetic Dataset

We used the identical semi-synthetic dataset generated by Melnychuk et al. (2022), which is based on real-world medical data from intensive care units, to validate our model with high-dimensional, long-range patient trajectories. As outlined in Melnychuk et al. (2022), this dataset builds on the MIMIC-III dataset and simulates patient trajectories with both endogenous and exogenous dependencies, taking treatment effects into account (Johnson et al., 2016). This setup allows us to control for confounding in our experiments. The use of semi-synthetic data is important here, as real-world data lacks ground-truth counterfactuals, which are necessary for evaluating our methods' performances. To make our manuscript self-sustained, we hereby summarize the setup elaborated in Causal Transformer (Melnychuk et al., 2022). Full details on the data generation process can be found in Appendix K Melnychuk et al. (2022).

Following (Melnychuk et al., 2022), we utilized MIMIC-extract (Wang et al., 2020) based on the MIMIC-III dataset (Johnson et al., 2016). The data were preprocessed with forward and backward imputation for missing values and standardization of continuous features. Our dataset included 25 time-varying signals and 3 static covariates (gender, ethnicity, age), yielding 44 total features ($d_w = 44$) after one-hot-encoding.

The simulation follows four main steps:

1. **Cohort Selection**
   1,000 patients whose ICU stays lasted between 20 and 100 hours are sampled .

2. **Untreated Outcomes**
   For each patient $i$, simulated $d_y$ untreated outcomes $\mathbf{Z}_t^{j,(i)}$ are simulated by combining:

- A B-spline term as an endogenous component
- Random function $g^{j,(i)}(t)$
- Exogenous covariate dependencies $f_Z^j(\mathbf{X_t}^{(i)})$
- Independent Gaussian noise $\epsilon_t \sim N(0, 0.005^2)$

$$\mathbf{Z}_t^{j,(i)} = \alpha_S^j \text{B-spline}(t) + \alpha_g^j g^{j,(i)}(t) + \alpha_f^j f_Z^j(\mathbf{X_t}^{(i)}) + \epsilon_t$$

3. **Treatment Assignment**
   We generated binary treatment indicators $\mathbf{A_t}^l$, $l = 1, ..., d_a$, based on previous outcomes and covariates, using a sigmoid function:

$$p_{\mathbf{A_t}^l} = \sigma(\gamma_A^l \bar{A}_{Tl}(\bar{Y}_{t-1}) + \gamma_X^l f_Y^l(X_t) + b_l)$$

$$\mathbf{A_t}^l \sim \text{Bernoulli}(p_{\mathbf{A_t}^l})$$

   Confounding is added by a subset of current time-varying covariates via a random function $f_Y^l(X_t)$, and $f_Y^l(\cdot)$ is sampled from an RFF approximation of a Gaussian process.

4. **Treatment Effects**
   In this step, treatments are applied to the initial untreated outcomes. We start by setting $\mathbf{Y_1} = \mathbf{Z_1}$, where each treatment $l$ influences an outcome $j$ with an immediate, maximum effect $\beta_{lj}$ after application. The treatment effect occurs within a time window from $t - w^l$ to $t$, with effect decreasing according to an inverse-square decay over time. The effect is also scaled by the treatment probability $p_{\mathbf{A_t^l}}$. When multiple treatments are involved, their combined effect is calculated by taking the minimum across all treatment impacts.

   The aggregated treatment effect is given by:

$$E^j(t) = \sum_{i=t-w^l}^{t} \frac{\min_{l=1,...,d_a} \mathbb{1}_{[\mathbf{A_i^l} = 1]} p_{\mathbf{A_i^l}} \beta_{lj}}{(w^l - i)^2}$$

5. **Combining Treatment Effects**
   We then add the simulated treatment effect $E^j(t)$ to the untreated outcome $Z_t^j$ to get the final outcome:

$$Y_t^j = Z_t^j + E^j(t)$$

6. **Dataset Generation**
   The semi-synthetic dataset was generated using the above framework. For the exact parameter values used in the simulation, please refer to the GitHub repository of Melnychuk et al. (2022). Following the setup in CT, we used the simulated three synthetic binary treatments ($d_a = 3$) and two synthetic outcomes ($d_y = 2$). We also use the identical setup and split the 1000-patient cohort into training, validation, and test sets, with a 60%/20%/20% split. For one-step-ahead prediction, all $2^3 = 8$ counterfactual outcomes were simulated. For multiple-step-ahead prediction, we sampled 10 random trajectories for each patient and time step, with $\tau_{\max} = 10$.

### E.2.1 Model Hyperparameters

Benchmark method hyperparameters and performance are sourced from the GitHub repository of Melnychuk et al. (2022).

Table 9: Model hyperparameters used for the semi-synthetic dataset.

| | |
|---|---|
| CT + SGA + RTM | batch size = 64, 
 lr = 0.01, 
 $\lambda = 0.0001$, 
 dropout rate = 0.1, 
 Adam |
| CRN + SGA + RTM | encoder batch size = 128, 
 encoder lr = 0.001, 
 encoder dropout rate = 0.1, 
 decoder batch size = 512, 
 decoder lr = 0.0001, 
 decoder dropout rate = 0.1 
 , $\lambda = 0.0001$, 
 Adam |

## F    Computational Analysis of SGA

While we focus on the theoretical analysis using Wasserstein distance, we recognize the necessity of computational studies due to its high computational cost. In particular, when the probability measures have at most n supports, the computational complexity of the Wasserstein distance is on the order of $O(n^3 log(n))$ (Pele & Werman, 2009). In high-dimensional settings (e.g., images with many pixels), the size of these supports can become very large, compounding the computational burden.

To this end, we employ three approaches to improve the computational efficiency: (i) an embedding network to reduce the dimension of the input and the sinkhorn algorithm (Altschuler et al., 2017) for estimating the Wasserstein distance. Through an embedding/representation learning network, high-dimensional inputs can be mapped into low-dimensional spaces, thus considerably reducing the computational cost. (ii) Sinkhorn algorithm for practicality: We use the Sinkhorn algorithm (Altschuler et al., 2017) to approximate the Wasserstein distance, as it provides a more tractable approach than directly solving the optimal transport problem. Although this introduces an approximation error, the regularization parameter $\epsilon$ allows us to balance accuracy and efficiency. Notably, the entropic version can achieve an approximate Wasserstein distance computation in $O(n^2)$ (Nguyen et al., 2022) (up to polynomial orders of approximation errors). (iii) Clustering is performed periodically, not every iteration, balancing performance with computational cost and stability (Details discussed in Appendix D).

## G    Discussions and Limitations

We address the critical challenge of counterfactual outcome estimation in time series by introducing two novel, synergistic approaches: Sub-treatment Group Alignment (SGA) and Random Temporal Masking (RTM). SGA tackles time-varying confounding at each time point by aligning fine-grained sub-treatment group distributions, leading to tighter counterfactual error bound and more effective deconfounding. Complementarily, RTM promotes temporal generalization and robust learning from historical patterns by randomly masking covariates, encouraging the model to preserve underlying causal relationships across time steps and rely less on potentially noisy contemporaneous time steps.

Our comprehensive experiments demonstrate that SGA and RTM are broadly applicable and significantly enhance existing state-of-the-art methods. While each approach individually improves performance, their synergistic combination consistently achieves SOTA performance on both synthetic and semi-synthetic benchmark datasets. Together, SGA and RTM offer a flexible and effective framework for improving causal inference from observational time series data.

In this work, we focus on synthetic and semi-synthetic datasets to enable controlled evaluation of counterfactual outcomes, where ground truth is available, which is a crucial factor for validating methodological performance. Moving forward, applying our method to real-world data is an important next step. In particular, we are actively exploring a case study on interventions in depressive phenotypes in animal models (e.g., varying treatment response depending on phenotype). This experimental setting allows regularly collected behavioral and physiological data, motivating the techniques proposed here. We believe this setting is a natural fit for

our framework, and will facilitate real-world validation while maintaining fine-grained observational control. We then plan to work with clinical observational studies to further apply the work.

