# OpenReview forum: "Synergizing Deconfounding and Temporal Generalization For Time-series Counterfactual Outcome Estimation"
_TMLR — Accepted by TMLR_

### Review · Reviewer_EXDr · 2026-02-07

**Summary Of Contributions:**

# Summary

This paper introduces two techniques to improve counterfactual time series forecasting, called subgroup alignment (SGA) and random temporal masking (RTM). SGA proceeds by aligning latent subgroups, with the claim that this provides tighter bounds on counterfactual error (though I have doubts about this claim, see below). RTM randomly masks covariates, with the hope of increasing robustness, motivated by masking in language modelling. An empirical study, including semi-synthetic data and data from a pharmakokinetics benchmark, show improved performance relative to baselines without SGA or RTM. Ablation studies show that both SGA and RTM are both important to success.

### Strengths

Counterfactual time series prediction is an important and difficult topic. The empirical results show strong performance gains over existing approaches, with the settings and ablation studies both showing independent and complementary benefits of SGA and RTM. The idea of exploiting subgroup structure makes sense, and being a modified loss over representations that can be dropped in to related work is a big plus.

### Weaknesses

**Interpretation of the Key Theorem.** I have some concerns regarding the main theorem for SGA, Theorem 4.2. It is claimed in several places that SGA results in a tighter bound (e.g., "we theoretically demonstrate that SGA optimizes a
tighter upper bound" in the abstract, "we propose to use the sub-treatment group structures to achieve tighter counterfactual error bound" on page 4). Instead, Theorem 4.2 says the resulting bound is "not much worse," actually proving a generically looser bound.

In particular, it says that if one bounds distances over subgroup components, the resulting bound is potentially tighter, and will not exceed a constant additive factor $\delta$ over the original bound; i.e., in the original bound, you replace term $A$ with term $B$, where $B \leq A + \delta$, which isn't necessarily a tighter bound. It's empirically shown that $\delta$ may be small in practice, and further that $B \leq A$, but this does not align well to the claims, in my opinion.

Additionally, it is stated that "under mild conditions," Theorem 4.2 holds. These mild conditions are not stated in the main manuscript, nor argued to be mild in the main manuscript.

**Technical Content is Lacking in the Main Submission.** There are a number of technical content or claims that are deferred to the appendices, and do not appear in the main manuscript. While this is not so unusual in a machine learning paper generally, the main content of this paper is quite short and some of the deferred content quite fundamental. Examples include
- The causal assumptions used within the potential outcomes framework (deferred on pg. 2 to Appendix A).
- The mathematical definition of the W_1 loss (deferred on pg. 4 to Appendix C).
- Full formal statements of theorems (and their assumptions) are deferred to appendices.
  - This is particularly striking in Theorem 4.2, where the definition of the key term in interpreting the theorem ($\epsilon$) is missing.

This is partially reasoned as a space constraint, but TMLR does not have a page limit, and the main submission is actually quite short for a TMLR paper (10 pages of main content, where the typical "regular submission" is up to 12 pages of main content).

**Audience:**

Yes

**Audience Explanation:**

The approach is simple, achieves strong performance, and addresses a timely and important problem. The resulting approach is thus very likely to be interesting to some of the machine learning audience.

**Broader Impact Concerns:**

N/A.

**Claims And Evidence:**

No

**Claims Explanation:**

Please see the above weaknesses. In particular, the interpretation of the key theorem is what influences my answer to this question the most. See also the issues of clarity in requested changes, below.

**Requested Changes:**

**Clarify Claims of the Main Theorem [Critical].** Please clarify the claims of the main theorem (see above in "Weaknesses").

**Improved Clarity [Critical]** There are some ambiguities in the current writing. I list a few below:
- On pg. 2, "thus, for each i, we observe"; what $i$ is indexing (presumably $N$?) is not written.
- In the theorems, $\epsilon_F^*$ is used but not introduced.
  - Then, in Theorem 4.2, $\epsilon_F$ (no star) is used. But actually, $\epsilon_F^*$ is used in the full statement of the theorem in the appendix.

**Include or Justify CT's Omission in Semi-Synthetic Data [Critical].** Currently, CT (which is used as a competitive baseline elsewhere) does not appear in Table 2. Please justify this, or include the relevant results.

**Reference Capitalization [Minor]** There are many capitalization errors in the references, both in venue (e.g., "Neural information processing systems" should be "Neural Information Processing Systems") and in titles (e.g., "Gaussian" should be capitalized). Please correct these.

---

> ### Author Response · Authors · 2026-03-02
> **Response to Reviewer EXDr**
>
> We sincerely thank the reviewer for your rigorous and detailed assessment of our work. Your feedback on the interpretation of Theorem 4.2 and the organization of the manuscript is invaluable. We appreciate the acknowledgment that our approach is simple, achieves strong performance, and addresses a timely problem.
>
> We address your critical concerns and requested changes below:
>
> **Mathematical Interpretation of Theorem 4.2**
>
> *Theorem 4.2 proves a bound B $\leq$ A + $\delta$). It does not necessarily prove a tighter bound.*
> > **Response:** We sincerely thank the reviewer for this precise mathematical observation. We fully agree that our sub-treatment group alignment does not guarantee a universally tighter bound than marginal alignment, as it is strictly tighter up to an additive term. We will revise the manuscript’s wording to explicitly reflect this mathematical reality. We also appreciate the opportunity to clarify the sense in which our bound is practically tighter and more effective for deconfounding.
> > * **Marginal vs. conditional tightness:** Standard methods (Theorem 4.1/Eq 2) rely on the marginal distribution distance (A). While A is a valid upper bound, it can be loose in heterogeneous populations because it does not take sub-group variance within consideration. Because marginal alignment seeks to match the overall global distributions of the treatment and control groups, it may inadvertently encourage the alignment of fundamentally dissimilar instances. In contrast, Theorem 4.2 establishes that minimizing the conditional discrepancy B offers a more targeted approach. By focusing alignment within locally similar sub-groups, the model navigates a more tractable optimization space, allowing for finer-grained control over counterfactual risk.
> > * **The additive term $\delta_c$** : The additive constant $\delta_c$ is $4\sqrt{\epsilon}$ is governed by **intra-cluster compactness** $\epsilon$, defined as: ${\max_{k \in [K]}}(\text{tr} (\Sigma_k^{0})) \leq \epsilon$ and ${\max_{k\in[K]}}(\text{tr} (\Sigma_k^{1})) \leq \epsilon$.} This constant is not an arbitrary penalty; it directly measures **the quality of the identified sub-treatment groups**. If the model identifies sub-groups with high homogeneity and small intra-cluster variance, $\epsilon$ becomes small, ensuring that the conditional alignment is an effective objective for deconfounding.
> > * **Empirical tightness (Figure 9):** The claim of tightness is supported by our empirical measurements of the bounds. As shown in Figure 9, we found that the SGA alignment loss (B) converges to consistently lower values than the marginal loss (A) during training. In other words, while $B \leq A + \delta$ is the theoretical relationship, we observe in practice that the value of B we optimize is smaller than the value of A optimized by the baseline method. To better reflect your point and strengthen the bridge between theory and practice, **we have now explicitly referenced and discussed this comparison in the main text (Remark 4.4, immediately following Theorem 4.2)**. This revision makes clear that the empirical reduction in Wasserstein discrepancy directly demonstrates a tighter constraint in practice, and it aligns with the observed performance improvements in our experiments.
>
> > **Revision:** To ensure mathematical precision, we have updated the text throughout the manuscript to **move away from the 'tighter bound' phrasing**. Instead, we explicitly state that the bound is theoretically tighter only up to an additive constant term, while emphasizing that it proves to be tighter in practice.
> >Specifically, we use the following description: *SGA optimizes a finer-grained conditional objective that is theoretically tighter than marginal alignment up to an additive constant term, while empirically yielding a tighter constraint on counterfactual risk in practice.*

---

> > ### Author Response · Authors · 2026-03-02
> > **Response to Reviewer EXDr**
> >
> > **Organization and technical content in the main manuscript (all changes have been highlighted in blue in the revised manuscript)**
> > > **Response:** We sincerely thank the reviewer for these suggestions, which will greatly improve the clarity and self-sufficiency of our paper.
> >
> > **Including Causal Transformer (CT) in Table 2.**
> > > **Response:** We **fully agree** with the reviewer that including the full CT model in the MIMIC-III experiments is necessary for a comprehensive evaluation. We initially did not include this baseline because its performance on this specific dataset is very similar to its ablated version, CT($\alpha=0$) (CT without marginal alignment), as reported by Melnychuk et al. (2022) [1]. This marginal difference suggests that the alignment loss used for deconfounding in standard CT yields limited benefits here. Instead, CT’s strong performance **on this specific dataset** is largely due to the transformer architecture effectively leveraging historical information. However, SGA remains highly effective in our other experimental setups.
> >
> > > **Revision:** Following the reviewer's suggestion, we agree that including this comparison makes our paper much more comprehensive. We have now included CT($\alpha=0$), the full CT, and our proposed method (CT+SGA+RTM) as shown below. We **have updated Table 2 in the revised manuscript** to include these results, ensuring a direct and complete comparison.
> >
> > |                                  | τ = 1 | τ = 2 | τ = 3 | τ = 4 | τ = 5 | τ = 6 | τ = 7 | τ = 8 | τ = 9 | τ = 10 |
> > |---------------------------- |-------|-------|-------|-------|-------|-------|-------|-------|-------|--------|
> > | MSMs                       | 0.37  | 0.57  | 0.74  | 0.88  | 1.14  | 1.95  | 3.44  | >10.0 | >10.0 | >10.0  |
> > | RMSNs                    | 0.24  | 0.47  | 0.60  | 0.70  | 0.78  | 0.84  | 0.89  | 0.94  | 0.97  | 1.00   |
> > | G-Net                       | 0.34  | 0.67  | 0.83  | 0.94  | 1.03  | 1.10  | 1.16  | 1.21  | 1.25  | 1.29   |
> > | CRN                         | 0.30  | 0.48  | 0.59  | 0.65  | 0.68  | 0.71  | 0.72  | 0.74  | 0.76  | 0.78   |
> > | CRN + SGA + RTM | 0.27  | 0.43  | 0.52  | 0.58  | 0.62  | 0.65  | 0.67  | 0.69  | 0.72  | 0.73   |
> > | CT (α = 0)                | 0.20  | 0.38  | 0.46  | 0.50  | 0.52  | 0.54  | 0.56  | 0.57  | 0.59  | 0.60   |
> > | CT                            | 0.21  | 0.38  | 0.46  | 0.50  | 0.53  | 0.54  | 0.55  | 0.57  | 0.58  | 0.59   |
> > | CT + SGA + RTM    | 0.21  | 0.38  | 0.44  | 0.50  | 0.52  | 0.52  | 0.56  | 0.57  | 0.58  | 0.58   |
> >
> >
> > >Reference:
> > >[1] Melnychuk, Valentyn, Dennis Frauen, and Stefan Feuerriegel. "Causal transformer for estimating counterfactual outcomes." International Conference on Machine Learning. PMLR, 2022.
> >
> > **Justification of mild conditions.**
> > > **Response:** We agree that explicitly clarifying the underlying assumptions is important for supporting the theoretical validity of our method. To address the concern regarding the unstated "mild conditions", we **have moved the full formal statements of Theorem 4.2 into the main text**. Furthermore, we have **added a dedicated paragraph in Remark 4.5 (Section 4.2)** that explicitly links these conditions to their theoretical motivations and empirical justifications:
> > > * **Assumption A1 (Gaussian sub-distributions in latent space):** We assume that sub-groups follow Gaussian distributions within the representation space. While raw data is rarely Gaussian, we justify this via the Central Limit Theorem, as deep encoders apply complex non-linear transformations that tend to produce more regular distributions (empirically supported by **Figure 8**). We will also highlight our **Sensitivity Analysis (Section 6.6)** showing that SGA remains robust even when using **k-means clustering**, which relies on fewer distributional assumptions.
> > > * **Distance assumption (A1, second part):** We assume corresponding sub-groups are closer than non-corresponding ones. We link this to **Table 6**, which provides direct numerical validation of this assumption.
> > > * **Assumption A2 (bounded covariance trace):** This defines $\epsilon$ as as the **intra-cluster compactness** ( ${\max_{k \in [K]}}(\text{tr} (\Sigma_k^{0})) \leq \epsilon$ and ${\max_{k\in[K]}}(\text{tr} (\Sigma_k^{1})) \leq \epsilon$}). We justify this as a desirable property of well-learned features, where identified sub-groups have high homogeneity and small intra-cluster variance. We link this to our **empirical analysis of the bound (Figure 9)**, which shows that in practice, our model learns representations where $\epsilon$ is sufficiently small to ensure a tight alignment.

---

> > > ### Author Response · Authors · 2026-03-02
> > > **Response to Reviewer EXDr**
> > >
> > > **Organization and technical content in the main manuscript (all changes have been highlighted in blue in the revised manuscript) - continued**
> > >
> > > **Main revision.**
> > > >We have expanded the main manuscript to ensure all fundamental causal and mathematical foundations are present in the main text:
> > > > * **Section 2:** We have moved the sequential assumptions (consistency, sequential positivity, and sequential ignorability) from the Appendix A to Section 2.
> > > > * **Section 4.1:** We included the formal mathematical definitions of the Wasserstein-1 distance, Lipschitz functions, and the factual/counterfactual losses.
> > > > * **Section 4.2:** We have moved the full formal statements of Theorem 4.2 into the main text.
> > >
> > > **Minor changes.**
> > > > * **Index clarification:** We thank the reviewer for noting the ambiguity on page 2; we have explicitly added that i∈{1,…,N} indexes the individuals (samples) in our dataset.
> > > > * Furthermore, we thank the reviewer for identifying the notation inconsistency; we have **standardized the use of  $\epsilon_F^{\star}$** throughout the manuscript.
> > > > * **Reference capitalization:** We thank the reviewer for this suggestion. We have conducted a comprehensive check of the BibTeX file to ensure that all venue names (e.g., "Advances in Neural Information Processing Systems") and proper terms within titles (e.g., "Gaussian", "Wasserstein") are correctly capitalized. We will also correct the bibliography style to ensure paper titles are not automatically lowercased.
> > >
> > > We believe these revisions address your concerns and improve the technical clarity of our submission. Thank you for your insightful reviews!

---

### Review · Reviewer_pUPR · 2026-02-09

**Summary Of Contributions:**

Sub-treatment Group Alignment (SGA): The paper introduces SGA, a novel fine-grained representation alignment strategy for time-series causal inference. Instead of aligning entire treatment groups in latent space (the coarse standard approach), SGA clusters individuals into sub-treatment groups (via treatment-agnostic clustering at each time step) and aligns corresponding sub-groups between treatments. The authors prove that this yields a tighter upper bound on counterfactual prediction error than conventional alignment, thereby more effectively mitigating time-varying confounding.

Random Temporal Masking (RTM): The paper proposes RTM, a data augmentation technique to improve temporal generalization. During training, covariates at random time points are replaced with Gaussian noise. This forces the model to rely on historical patterns rather than spurious short-term correlations, preserving underlying causal signals across time. RTM thus helps the model generalize to longer-horizon counterfactual predictions by reducing the level of reliance on immediate (potentially noisy) observations.

Synergistic Framework and Empirical Results: The two techniques are combined into a unified framework for time-series counterfactual outcome estimation. SGA addresses per-step deconfounding while RTM enhances robustness across time, and their mutually complementary roles lead to the state-of-the-art performance on benchmarks. The framework is shown to be architecture-agnostic: the authors integrate SGA+RTM into both an LSTM-based model (Counterfactual Recurrent Network) and a Transformer-based model (Causal Transformer), achieving consistent improvements on a synthetic pharmacokinetic/pharmacodynamic dataset and a semi-synthetic medical ICU dataset.

**Audience:**

Yes

**Audience Explanation:**

There is a community working on deconfounding in causal inference with time series data.

**Claims And Evidence:**

Yes

**Claims Explanation:**

The submission’s claims are well supported by both theoretical analysis and empirical evidence. Theoretically, the paper builds on established generalization bounds for counterfactual inference and shows that aligning sub-treatment groups yields a counterfactual error bound that is tighter (or no looser) than standard whole-group alignment. Theorem 4.2 formalizes this improvement under standard assumptions, extending prior results to a finer-grained alignment setting. While the analysis is presented for a static single-step case, the authors plausibly argue that improved per-step deconfounding translates to time-series settings. Overall, the theoretical justification is sound and consistent with the paper’s core claims.

Empirically, the evidence is strong and comprehensive. The proposed SGA+RTM framework is evaluated on both a fully synthetic PK-PD benchmark with controllable confounding and a semi-synthetic ICU dataset derived from real-world data. Across both datasets, and for both one-step-ahead and multi-step counterfactual predictions, SGA+RTM consistently outperforms a wide range of strong baselines, including MSMs, RMSNs, G-Net, CRN, and the Causal Transformer. The gains are especially pronounced under high confounding and for long-horizon predictions, supporting the claims that SGA improves deconfounding while RTM enhances temporal generalization. Ablation studies further isolate the individual contributions of SGA and RTM and show that their combination yields additive benefits. Additional analyses of masking strategies and clustering sensitivity indicate that the method is robust to key design choices. While direct measurements of representation balance (e.g., Wasserstein distance reductions) could further strengthen the empirical link to theory, the reported results already provide convincing support for the paper’s main claims.

**Requested Changes:**

1. Strengthen the empirical link to theory. To directly support the deconfounding claim, if possible, report quantitative measures of representation balance (e.g., Wasserstein-1 distance or another discrepancy metric) between treatment groups with vs. without SGA. In addition, relating counterfactual error to the theoretical bound on synthetic data would help demonstrate that the tighter bound achieved by SGA translates into improved empirical performance.

2. Establish statistical significance of improvements. While mean RMSE is reported, the paper should clarify whether these improvements are statistically significant. Including confidence intervals or statistical tests (e.g., paired comparisons against baselines) in the main results would strengthen confidence that the reported SOTA performance is robust, particularly where gains are modest (e.g., on semi-synthetic data).

---

> ### Author Response · Authors · 2026-03-02
> **Response to Reviewer pUPR**
>
> We sincerely thank the reviewer for the careful reading and for the positive assessment of both the theoretical and empirical contributions. We are greatly encouraged that you find the claims well supported and the evaluation comprehensive. We appreciate the insightful suggestions to further bridge the gap between theory and practice. Below we address the requested changes.
>
> **Strengthening the empirical link to theory**
> > Response: We thank the reviewer for this insightful suggestion. We **fully agree** that explicitly quantifying representation balance is essential for directly connecting our theoretical guarantees to empirical performance.
>
> >This analysis was **included in our original submission (Appendix E.1.3)**, where we quantitatively compare the alignment loss under marginal alignment versus SGA across varying confounding levels ($\gamma$). Specifically, we report the Wasserstein-1 discrepancy between treatment groups with and without SGA throughout training. As shown in Figure 9, the SGA alignment loss consistently converges to lower values than the marginal alignment loss, providing direct quantitative evidence that SGA achieves improved representation balance.
>
> >In addition, we provide empirical support for the key distance assumption (A1) underlying Theorem 4.2. In Table 6, we compare the Wasserstein-1 distances between corresponding sub-groups (paired distances) and non-corresponding sub-groups (non-paired distances) across treatment arms. We consistently observe that corresponding sub-groups are closer in the learned representation space, which supports the structural assumption required by SGA and further strengthens the connection between the theoretical analysis and empirical performance.
>
> >We apologize for not highlighting this connection more clearly in the main text of the original submission. To better reflect your point and strengthen the bridge between theory and practice, **we have now explicitly referenced and discussed this comparison in the main text (Remark 4.4, immediately following Theorem 4.2)**. This revision makes clear that the empirical reduction in Wasserstein discrepancy directly corresponds to the tighter theoretical bound and aligns with the observed improvements in our experiments.
>
> >We hope this clarification better demonstrates the empirical validation of our theoretical claims.
>
> **Establishing statistical significance**
> > We **fully agree** that explicitly reporting variability is important for supporting the robustness of our empirical findings.
>
> >The original submission included this analysis in Appendix E.1.6, where we report Normalized RMSE with error bars (mean $\pm$ standard deviation) for the challenging long-horizon prediction task at $\tau=6$ under six different confounding levels $\gamma$.
> That figure compares CT+SGA+RTM and CRN+SGA+RTM against strong baselines (MSMs, RMSNs, G-Net, CRN, CT), and shows that our methods consistently achieve competitive or superior performance while maintaining relatively tight variability as $\gamma$ increases. This indicates that the combination of SGA and RTM improves not only predictive accuracy but also stability under stronger time-varying confounding.
>
> >We acknowledge that this statistical evidence was not sufficiently highlighted in the main text. **To better reflect this point, we have now included the error-bar results in Section 6.1.** The revised figure extends Figure 3 by explicitly presenting mean $\pm$ standard deviation, making the robustness of our framework more transparent.

---

### Review · Reviewer_nqfo · 2026-02-16

**Summary Of Contributions:**

The paper targets the the estimation task of counterfactual outcomes in time-series data.
This paper introduces a framework with two components: Sub-treatment Group Alignment (SGA) and Random Temporal Masking (RTM).
The first component aims to align latent sub-treatment clusters using the Wasserstein distance to learn the sub-treatment group in the hidden space. The second component randomly masks input covariates with Gaussian noise during training to improve generalization from historical patterns. The proposed framework is integrated into existing representation learning architectures like Counterfactual Recurrent Networks and Causal Transformers and show outperformed results in experiments.

**Audience:**

Yes

**Audience Explanation:**

The TMLR audience would be interested in causal inference, time-series counterfactual.

**Broader Impact Concerns:**

No concerns

**Claims And Evidence:**

No

**Claims Explanation:**

1. The alignment objective in Sec 4.2 lacks sufficient intuitive explanation, as it is not immediately clear why the specific formulation in Equation 3 improves the generalization bound or what drives the design of minimizing W1 distance in SGA.

2. In Sec 5, there is a disconnect between the narrative claims for temporal generalization and the mathematical formulation, as the objective function in Sec 5 does not explicitly model historical patterns.

**Requested Changes:**

1. The manuscript heavily relies on the appendix for critical information (including assumptions, mathematical definitions, proofs, and implementation details), which significantly disrupts the reading flow and creates gaps in the primary body. Please adjust the organization.

2. The paper lacks a generative diagram or structural causal model visualization early in the text to clearly illustrate the underlying data generation mechanism and the assumed confounding structure.

---

> ### Author Response · Authors · 2026-03-02
> **Response to Reviewer nqfo**
>
> We thank the reviewer for your thoughtful feedback. Regarding your questions and concerns, we have listed our answers below.
>
> **Intuition and motivation for the SGA objective (Eq. 3)**
>
> >**Response**: Our method builds on the standard counterfactual generalization bound, where the counterfactual error depends on both factual loss and the distributional discrepancy between treatment groups in representation space [1].
> > The key limitation of standard representation-balancing methods is that they align marginal treatment distributions, implicitly assuming homogeneous treatment groups. In time-series settings, however, treatment arms often consist of heterogeneous sub-populations whose responses differ substantially. Aligning only marginal distributions may therefore leave substantial residual imbalance.
>
> > **Revision**: We have **added the following intuition and revised Section 4.2 accordingly**.
>
> > *We provide an intuition here. In many real-world settings, each treatment group is not homogeneous but instead forms a mixture of latent sub-populations. For example, in medical studies, patients may naturally form sub-groups **before** the beginning of experiments, based on latent variables such as demographic characteristics or genetic factors. Consider a scenario where patients are sub-grouped according to age (e.g., children, adults, seniors), gender, or genetic markers that influence their response to treatment. Even though these patients receive different treatments, the underlying characteristics defining the sub-groups are consistent across treatment groups. Marginal alignment minimizes the Wasserstein distance between the overall treatment distributions, but this coarse alignment can still permit substantial mismatch between corresponding latent sub-populations. In contrast, SGA decomposes each treatment distribution into sub-treatment components and aligns them pairwise. Intuitively, instead of forcing two heterogeneous clouds of representations to overlap globally, SGA aligns them locally at the sub-population level, leading to more effective deconfounding.*
>
> >Reference:
> >[1] Shalit, Uri, Fredrik D. Johansson, and David Sontag. "Estimating individual treatment effect: generalization bounds and algorithms." International conference on machine learning. PMLR, 2017.
>
> **Relationship between RTM formulation and temporal generalization.**
> > **Response**: We appreciate the chance to clarify how RTM relates to temporal generalization. We clarify that RTM is intentionally designed as a **training regularizer**, not a **modeling objective**.
>
> >RTM masks a subset of current covariates during training, forcing the model to rely on historical information rather than potentially spurious contemporaneous correlations.
> Empirical evidence supports this mechanism:
> >* **Ablation results in Table 3** show RTM improves performance especially for longer prediction horizons and stronger confounding.
> >* **Attention analysis in Section 6.5** demonstrates that RTM shifts the model’s focus toward historical time steps, with over 99% of its attention to previous time-points. This distributed attention across historical data demonstrates e!ective leveraging of past information.
>
> >To further show RTM’s advantage in temporal generalization, we have run additional longer-sequence experiments on the most challenging high-confounding setting, extending the prediction horizon up to $\tau$=20. The results shows a monotonic gain of RTM for temporal generalization.
> |                 |   CT  |  CT + RTM | Diff |
> |-|-|-|-|
> | $\tau=8$  |5.7      |2.1             | 3.6  |
> | $\tau=10$|  6.2   |      2.3        | 3.9  |
> | $\tau=12$|  6.6   |      2.5        | 4.1 |
> | $\tau=14$|   7.0      |    2.7          | 4.3 |
> | $\tau=16$|   7.3      |    2.8          | 4.5  |
> | $\tau=20$|   7.6      |    3.0          | 4.6   |
>
> > **Revision**: In addition, we have **revised Section 5 and added this clarification on the role of RTM in temporal generalization**.
>
> > *It is important to emphasize that RTM does not explicitly model temporal dynamics; rather, it functions as a training-time regularization mechanism. By randomly masking covariates at selected time steps with Gaussian noise, RTM reduces the availability of contemporaneous information during training. Consequently, the model must rely more heavily on historical observations when predicting future outcomes. This encourages the learning of representations that capture dependencies stable across time rather than spurious correlations localized to a single time step. In settings where current covariates are strongly correlated with factual outcomes, models without RTM may overfit to these contemporaneous signals, potentially impairing long-horizon counterfactual prediction. RTM mitigates this tendency by regularizing the hypothesis space and promoting temporal robustness. Empirically, this behavior is reflected in improved long-horizon performance and increased attention to past time steps.*

---

> > ### Author Response · Authors · 2026-03-02
> > **Response to Reviewer nqfo**
> >
> > **In response to the requested changes on manuscript organization**
> >
> > **Reliance on the appendix.**
> > > **Response:** Thank you for your constructive feedback on the reading flow. To make the paper self-contained:
> > > * **Revision 1**: We have moved the Sequential Causal Assumptions (A1–A3) to Section 2 and the definition of Wasserstein Distance to Section 4.1
> > > * **Revision 2**: We have moved the Lipschitz constants and Factual/Counterfactual loss definitions to Section 4.1 to ensure the theoretical motivation for SGA is fully understandable without the Appendix.
> >
> > **Generative diagram or structural causal model visualization early in the text.**
> > > **Response**: Thank you so much for your suggestion! We **have moved the Causal Directed Acyclic Graphs (DAGs) (Figure 7 in Appendix) to Section 2**. This will provide an immediate visual summary of the assumed data-generation mechanism and the challenges of time-varying confounding.
> >
> > **Summary of revisions (all changes have been highlighted in blue in the revised manuscript):**
> > > 1. **Section 2:** Integrated Causal Directed Acyclic Graphs (DAGs) to visualize the confounding structure and moved the core potential outcomes assumptions (A1–A3) from the Appendix to the main text.
> > > 2. **Section 4.1:** Included the formal mathematical definitions of the Wasserstein-1 distance, Lipschitz functions, and the factual/counterfactual losses.
> > > 3. **Section 4.2:** Added the full formal statement of Theorem 4.2 and expanded the intuitive text explaining the theoretical advantage of conditional (fine-grained) over marginal (coarse) alignment.
> > > 4. **Section 5:** Elaborated on how RTM serves as a stochastic training-time regularization mechanism that implicitly forces the model to prioritize historical causal patterns.

---

### Author Response · Authors · 2026-03-02
**Manuscript has been updated**

Dear Reviewers,

We sincerely thank you for your time and effort in reviewing our paper. We appreciate the rigorous feedback, which has helped us significantly improve the clarity, theoretical self-sufficiency, and empirical robustness of our work.

We have replied to each reviewer’s comments individually and uploaded an updated version of our manuscript. **For your convenience, we have highlighted all changes in blue**. A summary of the key revisions follows below:
* **Section 2** (Problem Formulation): We have moved the **Causal Directed Acyclic Graphs (DAGs)** and the core **sequential causal assumptions (A1–A3)** from the Appendix to the main text to immediately clarify the data generation mechanism. We also clarified the indexing notation.
* **Section 4.1**  (Theoretical Background): We moved the formal mathematical definitions of the **Wasserstein-1 distance, Lipschitz functions, and factual/counterfactual losses** from the Appendix to the main text to ensure the theoretical motivation is self-contained.
* **Section 4.2** (SGA Theoretical Analysis):
  * We moved the **full formal statement of Theorem 4.2** into the main text.
  * We added **Remark 4.4** to explicitly discuss the tightness of the bound, clarifying that SGA is theoretically tighter up to an additive constant but yields a strictly tighter constraint in practice.
  * We added **Remark 4.5** to justify the mild conditions and linked them to our empirical findings.
  * We revised the manuscript to explicitly state that the bound is **theoretically tighter up to an additive constant term**, while **empirically yielding a tighter** constraint on counterfactual risk.
* **Section 5 (RTM)** We revised the text to clarify that **RTM serves as a training-time regularization mechanism** rather than a  modeling objective , explaining how it enforces reliance on historical patterns.
* **Section 6 (Experiments)**: We updated **Table 2** to include the **Causal Transformer (CT) and CT($\alpha=0$)** baselines as suggested. We also added **error bars** (mean $\pm$ std) to the results to demonstrate statistical stability.

Please feel free to post more comments or questions if our answers do not fully address your concerns.

Best regards,
Authors of Submission 6899

---

> ### Author Response · Authors · 2026-03-13
> **Follow-Up on Responses and Revised Manuscript**
>
> Dear Reviewers,
>
> Thank you again for your time and consideration!
>
> Following up on our responses and the submission of our revised manuscript, we wanted to check in to see if our updates and clarifications have addressed your questions and concerns.
>
> As the discussion period progresses, please let us know if there are any additional comments or questions that we can address. We would be more than happy to provide any further clarification!
>
> Thank you again for your time and feedback.
>
> Best regards,
> Authors of Submission 6899

---

> > ### Author Response · Authors · 2026-04-04
> > **Follow-Up**
> >
> > Dear Reviewers,
> >
> > Thank you again for your time and consideration! We wanted to kindly follow up regarding our revised manuscript (submission 6899).
> >
> > We truly appreciate the thoughtful feedback you provided. Following your suggestions, we submitted a revised version along with detailed responses to each of your comments.
> >
> > We completely understand how busy things can be; we simply wanted to check in to see if our revisions have addressed your concerns, or if there is anything else we can clarify or improve. We would be more than happy to provide any additional information that might be helpful.
> >
> > Thank you again for your time and feedback.
> >
> > Warm regards,
> > Authors of Submission 6899

---

### Decision · Action_Editor_mCKW · 2026-04-14

**Recommendation:** Accept with minor revision

**Additional Comments:**

As discussed above, the empirical evidence is present, while the framing of the theoretical contribution is unjustified in its current form.

However, this is primarily a matter of framing and can be resolved in a minor revision.
This revision should include the following changes:

- A rephrasing of the claim of a tighter bound up to additive constants, to acknowledge a looser bound whose tightness is controlled by $\delta_c$.
- A short justification/discussion paragraph in the main paper explaining the validity of the assumptions for Theorem 4.2.

No changes to the method or empirical evaluation are needed.

**Audience:**

Yes

**Audience Explanation:**

Counterfactual outcome estimation from temporal data is a topic of ongoing research and of interest to parts of the TMLR community. All three reviewers agree on its relevance to TMLR's audience, and I share their opinion.

**Claims And Evidence:**

No

**Claims Explanation:**

While all reviewers agree on the empirical evidence, I share reviewer EXDr's reservation about Theorem 4.2.
The repeated claim of providing a tighter bound, up to an additive constant, is an incorrect formulation.
What the authors derive is a looser bound whose looseness is controlled by $\delta_c$.

---

> ### Author Response · Authors · 2026-05-14
>
> Dear Dr. Haussmann,
>
> We greatly appreciate your decision to accept our paper for publication in TMLR and your recognition of the significance of our contributions to the TMLR community!
>
> We have submitted the final camera-ready version of our paper, including a link to the code repository containing our model implementation and the experiments discussed.
>
> We have addressed your requested edits in the following ways:
>
> **Rephrasing the "tighter bound up to additive constants" claim.**
> > * Following your suggestion, we have heavily revised Remark 4.4 to **explicitly acknowledge** that the SGA bound can be looser than the marginal-alignment bound by at most the additive constant $\delta_c$​, with its tightness controlled by$\delta_c$. We retain the note that $\delta_c$ is small in practice, with Appendix E.1.3 providing the supporting empirical evidence.
> > * We have also propagated this framing **consistently across the manuscript**, by updating the abstract, the introduction, the opening of Section 4.2, and the conclusion so that every reference to the theoretical guarantee **explicitly notes** that the SGA bound is theoretically tighter **up to an additive constant term $\delta_c$.**
>
> **Justification of the assumptions for Theorem 4.2.**
> > * We have added Remark 4.5 in the main paper, which states the assumptions of Theorem 4.2 and discusses their validity, with detailed justifications deferred to Appendix C.
> > * The expanded discussion in Appendix C now covers: (i) why Gaussianity of sub-distributions in the learned representation space is plausible through deep-encoder regularization, the empirical histograms in Figure 8, and SGA's robustness to clustering algorithm choice as shown in Section 6.6; (ii) empirical validation of the distance assumption $W_1(P_{\Phi,k}^{0}, P_{\Phi,k}^{1}) \leq W_1(P_{\Phi,k}^{0}, P_{\Phi,k^{\prime}}^{1})$ for $k \neq k^{\prime}$ via Table 6, which reports paired vs. non-paired Wasserstein distances across multiple timepoints and treatment-arm pairs in the fully-synthetic dataset; and (iii) interpretation of the bounded-covariance-trace condition as a natural property of well-learned compact representations.
>
> We appreciate your detailed comments and believe these edits strengthen the clarity and rigor of our paper's theoretical framing. Please feel free to leave a comment if you require any further information from us.
>
> Thank you once again for your support and thoughtful review!
>
> Best regards,
> Authors of Submission 6899